# Dynamic modelling of weathering rates – the benefit over steady-state modelling

Veronika Kronnäs[1], Cecilia Akselsson[1], Salim Belyazid[2]

[1]Department of Physical Geography and Ecosystem Science, Lund University, 223 62 Lund, Sweden
[2]Department of Physical Geography, Stockholm University, 106 91 Stockholm, Sweden

*Correspondence to*: Veronika Kronnäs (veronika.kronnas@nateko.lu.se)

**Abstract.** Weathering rates are of considerable importance in estimating the acidification sensitivity and recovery capacity of soil, and are thus important in the assessment of the sustainability of forestry in a time of changing climate and growing demands for forestry products. In this study, we modelled rates of weathering in mineral soil at two forested sites in southern Sweden included in a monitoring network, using two models. The aims were to determine whether the dynamic model

ForSAFE gives comparable weathering rates as the steady-state model PROFILE, and whether the ForSAFE model provided believable and useful extra information on the response of weathering to changes in acidification load, climate change and land use.

The average weathering rates calculated with ForSAFE were very similar to those calculated with PROFILE for the two modelled sites. The differences between the models regarding the weathering of certain soil layers seemed to be due mainly

to differences in calculated soil moisture. The weathering rates provided by ForSAFE vary seasonally with temperature and soil moisture, as well as on longer time scales, depending on environmental changes. Long-term variations due to environmental changes can be seen in the ForSAFE results, for example: the weathering of silicate minerals is suppressed under acidified conditions due to elevated aluminium concentration in the soil, whereas the weathering of apatite is accelerated by acidification. The weathering of both silicates and apatite is predicted to be enhanced by increasing

temperature during the 21st century. In this part of southern Sweden, yearly precipitation is assumed to be similar to today's level during the next forest rotation, but with more precipitation in winter and spring and less in summer, which leads to somewhat drier soils in summer, but still increased weathering. In parts of Sweden with bigger projected decrease in soil moisture, weathering might not increase despite increasing temperature.

These results show that the dynamic ForSAFE model can be used for weathering rate calculations and that it gives average

results comparable to those from the PROFILE model. However, dynamic modelling provides extra information on the variation in weathering rates with time, and offers much better possibilities for scenario modelling.

## 1 Introduction

Most parts of Sweden are covered with glacial till, composed largely of slowly weathering minerals of granitic origin, the type of rugged landscape with mostly very shallow soil depth described in Krabbendam and Bradwell (2014). This makes

both soils and lakes sensitive to acidification. Two thirds of Sweden is covered by boreal and northern temperate forests, mostly consisting of Norway spruce and Scots pine, together with birch and a few other deciduous trees. Forests are one of Sweden's most important natural resources, and are used for timber (32 million $m^3 y^{-1}$ in 2013, Christiansen, 2014), pulp wood (31 million $m^3 y^{-1}$ in 2013) and biomass for energy production (6 million $m^3 y^{-1}$ in 2013). The last is especially

important due to the need to replace fossil fuels with renewable sources of energy (Chu and Majumdar, 2012). Forests and their soil also determine the water quality of most lakes and streams, since the catchments of most lakes are forested and surface water is filtered through forest soils.

During the 1960s, lakes in Scandinavia became increasingly acidified (Odén, 1968). The cause of this was found to be air pollution in the form of atmospheric sulphur and nitrogen (Overrein, 1972), much of it from fossil fuel combustion. The

regions most severely affected were those with high deposition of acidifying substances on shallow soils containing base cation poor minerals with low weathering rates and release of base cations (i.e. calcium, magnesium, potassium and sodium) (Galloway et al., 1983). In 1979, the Convention on Long-Range Transboundary Air Pollution (CLRTAP) was formulated by the United Nations Economic Commission for Europe (UNECE). CLRTAP was extended by the addition of several protocols for the mitigation of air pollutants, where participating countries were urged to submit data on emissions of

pollutants and ecosystem sensitivity. A need thus emerged for ways of assessing ecosystem sensitivity, and different methods of estimating critical loads of acidity for sulphur and nitrogen for forest and lake ecosystems were developed (Sverdrup and Warfvinge, 1995). One of these was the PROFILE model, developed by researchers at Lund University during the 1990s (Sverdrup and Warfvinge, 1993; Sverdrup et al., 2005).

CLRTAP led to a considerable reduction in the emission of acidifying pollution, and lakes and soils in large parts of

acidified areas in Europe slowly started to recover (Engardt et al., 2017; Garmo et al., 2014; Johnson et al., 2018). However, acidifying pollution is still a large and increasing problem in some parts of the world, for example, Southeast Asia (Cho et al., 2016). Forestry is also a potentially acidifying practice, as buffering base cations are removed during harvest (Farley and Werritty, 1989; Akselsson et al., 2016; Zetterberg et al., 2013). Furthermore, as the demand for forest products is growing, while both climate conditions and atmospheric deposition are changing, there is an increasing need to evaluate the sensitivity

of forest soils and the weathering of base cations in greater detail, as an aid in forestry planning and regulation. The dynamic ecosystem model ForSAFE (Wallman et al., 2005; Belyazid et al., 2006), which consists of a dynamic development of the PROFILE model, together with models for tree growth and decomposition, has the potential to do this.

The aims of this study were:

•       to investigate whether ForSAFE gives comparable weathering rates to those estimated with the PROFILE model,

and to explain the results based on differences in the formulation of the models, and

•       to investigate the seasonal, inter-annual and decadal weathering dynamics provided by ForSAFE for different scenarios, representing important ecological issues: acidification, climate change and nutrient removal through land use.

## 2 Methods

The PROFILE and ForSAFE models were applied to two spruce forest sites in southernmost Sweden, Västra Torup and Hissmossa, included in the Swedish Throughfall Monitoring Network (SWETHRO) (Pihl Karlsson et al., 2011). Different scenarios for the input parameters were modelled with ForSAFE. ForSAFE-modelled weathering for the base scenario was averaged over the 21st century forest rotation and compared with PROFILE-modelled weathering. The weathering rates from the different scenarios from the ForSAFE model were examined in detail.

### 2.1 PROFILE

The PROFILE model is a steady-state mechanistic biogeochemistry model, developed at Lund University in the 1990s (Sverdrup and Warfvinge, 1993; Warfvinge and Sverdrup, 1995). It has been widely used for calculations of critical loads of acidification, weathering as an aid to improving the sustainability of forestry in Europe (including Iceland with its very different mineralogy), North America and East Asia, and has even been applied to agricultural land (Akselsson et al., 2016; Erlandsson et al., 2016; Phelan et al., 2014; Fumoto et al., 2001; Holmqvist et al., 2003; Stendahl et al., 2013). The ecosystem in PROFILE is represented by a soil profile divided into layers, each with its own chemical and physical properties, to which water, nutrients and pollutants are added via atmospheric deposition and litterfall from trees, and from which water, nutrients and pollutants are removed via uptake by trees and downward leaching. Chemical equilibrium reactions and weathering take place in the soil profile. Weathering is modelled using transition state theory, and the parameters affecting it are soil temperature, soil moisture, mineralogy, soil texture, expressed as the exposed mineral surface area, soil density, and the concentrations of $H^+$, organic ligands and carbon dioxide, as well as the concentrations of inhibitors: base cations (Ca, Mg, K and Na), $Al^{3+}$ (products of the weathering reaction) and organic acids.

### 2.2 ForSAFE

The ForSAFE model consists of a dynamic development, SAFE, of the PROFILE model (Alveteg et al., 1995; Martinsson et al., 2005), together with the DECOMP model of the decomposition of soil organic matter (Wallman et al., 2006; Walse et al., 1998), the PnET model of tree growth (Aber and Federer, 1992) and the hydrological PULSE model (Lindström and Gardelin, 1992). ForSAFE was developed to better model the process of recovery from acidification and the effects on ecosystems of forestry and climate change, with dynamic feedbacks between soil chemistry and forest growth. Many parameters used as input data in the PROFILE model are modelled by the ForSAFE model. These include runoff, soil moisture, decomposition of litter and the uptake of nutrients by trees. The model is being continuously developed (Belyazid et al., 2011; Phelan et al., 2016; Zanchi et al., 2014; Yu et al., 2016; Rizzetto et al., 2016; Gaudio et al., 2015). In this study, a ForSAFE version with monthly time steps was used.

## 2.3 Site descriptions

The characteristics of the two SWETHRO sites, at Västra Torup and Hissmossa, are presented in Table 1 and Table 2. SWETHRO is a Swedish network started in the 1980's to monitor deposition of acidifying substances to Swedish managed forest and how the forest and forest soil is affected by the deposition. Each site consists of a 30 m x 30 m square plot in a forest stand, where throughfall deposition is measured every month, and soil water chemistry parameters are measured with lysimeters at a depth of 50 cm three times per year; at Västra Torup since 1996, and at Hissmossa since 2010. Open field deposition is measured near the stands. Soil chemistry, texture and other properties as well as forest parameters have been measured previously (Tables 1 and 2).

Västra Torup has previously been modelled by Belyazid et al. (2006) with an earlier version of the ForSAFE model, using less detailed input data. Zanchi et al. (2014) have also modelled this site using the same version of ForSAFE as in the present study, as well as most of the input data, with the aim of describing changes in forest ecosystem services in a changing climate.

The forest at Västra Torup was clear cut in 2010, and the site at Hissmossa, 5 km to the north, was introduced into SWETHRO as a replacement site. Hissmossa has previously been modelled with ForSAFE, with the aim of explaining why this site shows continuously elevated concentrations of nitrate in soil water, while Västra Torup did not, prior to clear cutting (Olofsson et al., manuscript). Hissmossa has courser, very sandy soil. Both soils are high in quartz and feldspars. Both sites are highly productive sites for Norway spruce, but was probably grazing lands up to the beginning of the 20[th] century. The soils are assessed as transition types.

The soil parameters used in the modelling are given in Table 3. Values of the field capacity and wilting point were calculated using the equations given by Balland et al. (2008). Mineral content was calculated from total soil chemistry data using A2M, a mathematical model that uses total chemistry of the soil samples to come up with possible mineral compositions (Posch and Kurz, 2007). For the uppermost, organic layers, mineralogy and texture from the second layers were used, since there are no texture analyses for the organic layers and the total chemistry analyses of the organic layers include the ash of the organic matter. The soil moisture input value for PROFILE is an estimated site specific value based on observations at the sites. In this case the soil moisture value is equal at both sites: 0.2 $m^3_{soil\ water\ volume}m^{-3}_{soil\ volume}$ for all layers. The fraction of stones in the soils is also estimated at the time of the soil sampling.

## 2.4 Scenarios and time series of driver parameters

ForSAFE uses time series of climate parameters, forest management and the deposition of atmospheric pollutants and base cations to the site. A set of these time series, from 1900 to 2100, is here called a scenario. The purpose of the different scenarios used in this study is to investigate how ForSAFE-modelled weathering rates responded to changes in the driving parameters. Thus, the scenarios used consist of a base scenario (BSC), four scenarios in which one aspect of the environment differs from the BSC scenario and a background scenario (BGR) without forestry, acidification and climate change.

The BSC scenario represents the actual drivers at the sites from 1900 to today, followed by a reasonably realistic future to the year 2100 with regards to forestry management, climate and deposition. This scenario has been used by Zanchi et al. (2014), and Olofsson et al. (manuscript). The future climate is based on a high-$CO_2$ emission scenario (SRESA2, modelled with ECHAM5: Nakićenović et al., 2000; Roeckner et al., 2006), with an approximately exponentially increasing temperature during the 21st century. Annual precipitation is almost unaffected by the climate change in this scenario for this part of Sweden up to 2080, after which it increases, but only by about 8%. The distribution of precipitation during the year changes after 2050, with more precipitation during winter and spring and less during summer. Past and future forest management of the sites in the BSC scenario is based on normal, but not intensively, managed forestry in Sweden today, with two thinnings (at approximately 30 and 45 years after planting) and clear cuttings approximately every 70 years, where only stem wood is removed. The deposition of pollutants and base cations is based on data from the EMEP programme (Simpson et al., 2012), with SOx-deposition peaking in 1970 and decreasing sharply after that and nitrogen deposition peaking in 1985 with a smaller decrease after that. Future deposition is assumed to be constant after 2020.

Five scenarios were compared with the BSC scenario, where climate, deposition or forest management were changed (for the whole or part of the period 1900 - 2100), while the other input parameters were as in the BSC scenario. The scenarios were:

• BSC: Base scenario, described above.

• NFO: No forestry: no thinning or clear cutting between 1900 and 2100. Deposition and climate change as in BSC.

• WTH: Whole-tree harvest at clear cutting and thinning from 2010. Deposition and climate change as in BSC.

• NAC: No acidification: no increase in acidifying deposition after 1900. Forestry and climate change as in BSC.

• NCC: No climate change: no increase in temperature between 1900 and 2100. Forestry and deposition as in BSC.

• BGR: Background: no clear cutting or thinning, no increase in acidifying deposition and no climate change.

## 3 Results

### 3.1 Weathering rates from PROFILE and ForSAFE

The total weathering rates obtained with ForSAFE with the BSC scenario, averaged over a forest rotation, were similar to the weathering rates obtained with PROFILE for all soil layers and modelled elements, and almost equal for many of them (Figure 1). At Västra Torup, the total annual weathering rate of the base cations (Ca, Mg, K and Na) in the root zone (organic layer plus the 50 uppermost cm of the mineral soil, L1-L5) was 115 meq $m^{-3}$ $y^{-1}$ on average, according to ForSAFE (varying for different months between 51 meq $m^{-3}$ $y^{-1}$ and 260 meq $m^{-3}$ $y^{-1}$), and 106 meq $m^{-3}$ $y^{-1}$ according to PROFILE. At Hissmossa, the total weathering rate of base cations in the root zone (L1-L4) estimated with ForSAFE was 38 meq $m^{-3}$ $y^{-1}$ (varying from 16 meq $m^{-3}$ $y^{-1}$ to 86 meq $m^{-3}$ $y^{-1}$) and 45 meq $m^{-3}$ $y^{-1}$ according to PROFILE.

The estimated weathering rate of base cations is lower at Hissmossa than that at Västra Torup according to both models. This is due to the coarser soil texture at Hissmossa, leading to a significantly lower exposed mineral surface area. Also, according to field measurements, Hissmossa has a more acid soil solution than Västra Torup, with twice the concentration of inorganic

aluminium at Västra Torup. Dissolved inorganic aluminium, a product of the weathering of silicate minerals, inhibits the weathering of silicate minerals. The rotation periods at Västra Torup and Hissmossa are not the same, so average rates for the forest rotation are not directly comparable since climate changes during the period. The differences in weathering rates between the sites are much larger than the changes in rates because of climate change in the two to three decades differing in

rotation period.

Differences in the weathering rates predicted by the two models are greater for soil layers where the differences between the values of soil moisture are higher between the two models (Figure 2), i.e. in the organic layers (where weathering is very small, due to very small mineral mass) and in L4 in Hissmossa. The input value for PROFILE was $0.2 \text{ m}^3_{\text{soil water volume}} \text{ m}^{-3}_{\text{soil}}$ $_{\text{volume}}$ for all layers at both these sites. The soil moisture is dynamically modelled in ForSAFE, with average values close to

the defined field capacity for the respective layers (Table 3). The average soil moisture at Västra Torup, for the forest rotation 2011 - 2080, was 0.18 - 0.21 in the mineral layers and 0.29 in the thin organic upper layer. In the sandy soil at Hissmossa the average soil moisture in ForSAFE (for the forest rotation 2041 - 2100) was 0.13 - 0.18 in the mineral soil layers and 0.4 in the organic soil layer. The difference between the value of soil moisture used in PROFILE and that calculated by ForSAFE is thus greater at Hissmossa, and the differences in weathering rates between the two models are thus

also greater at Hissmossa than at Västra Torup.

### 3.2 Seasonal, yearly and decadal variation in weathering rates from ForSAFE

The weathering rates obtained with ForSAFE vary seasonally with temperature and soil moisture, as well as on longer time scales, depending, for example, on forest stage, the acidification status of the soil and the climate (Figure 3). On the seasonal scale, weathering is lowest in winter and highest in the warmest period of summer, unless the soil is too dry. Weathering

rates during the warmest month of the year are typically 3 to 4 times higher than during the coldest month, except for Ca and P, where weathering in the warmest month is 5 to 8 times higher than in the coldest month. On longer time scales, the yearly average weathering rates can vary by a factor of two during a forest rotation.

### 3.3 Effect of forestry on weathering

Thinning and clear cutting at Västra Torup increased the weathering of base cations by 9 % in the future forest rotation

(2011 - 2080) in the BSC scenario, compared to the NFO scenario with no clear cutting or thinning (Figure 4). Whole-tree harvesting in the WTH scenario increased the weathering by a further one percent. At Hissmossa the increase in weathering between the NFO scenario and the BSC scenario was 14 % for the forest rotation between 2041 and 2100, with a further increase of 2 % for the WTH scenario. The difference in weathering between scenarios occurs during the first half of the forest rotation.

### 3.4 Effect of acidification on weathering

In ForSAFE, the weathering of silicate minerals is decreased by the acidified conditions in the soils during the second half of the 20th century in the BSC scenario, whereas the weathering of the only P-containing mineral, apatite, is enhanced (Figure 5). The effect of acidification on weathering is smaller than the effects of temperature and soil moisture. For the forest rotation 1941 - 2010 in Västra Torup, the weathering of base cations was 11 % lower in the BSC scenario than in the non-acidification NAC scenario, while the P weathering was 11 % higher. At Hissmossa, for the forest rotation 1973 - 2040 (i.e., mostly after the most acidified period), the weathering of base cations was 6 % lower and the weathering of P 17 % higher in the BSC scenario than in the NAC scenario.

### 3.5 Effect of climate change on weathering

Temperature has a considerable effect on weathering rates. In the BSC scenario, the yearly average temperature increased from 7˚C in the 1990s to 11˚C in the 2090s. This leads to an increase in ForSAFE weathering rates of the base cations of 7 % per degree increase in temperature. The increase in temperature is greatest in winter (6˚C difference between 1900 - 1930 and 2080 - 2100) and smallest in summer (4˚C difference between 1900 - 1930 and 2080 - 2100). In Hissmossa, the weathering rates of Ca in L4 are 44 % to 49 % higher in 2080 - 2100 in the BSC scenario than in the constant climate scenario, NCC, for all seasons (Figure 6).

### 3.6 Overall effect of forestry, acidification and climate change

The overall effect of human practices on weathering rates, as in the BSC scenario: forestry, historical acidification and climate change, is positive, compared to the background scenario, BGR. Climate change and forestry have a positive effect on silicate weathering, while acidification has a negative effect, but not of such a magnitude that it cancels out the first two. For apatite weathering, the combined effect of climate change, forestry and decreasing acidification is an increase of the weathering in the future, especially for newly planted forest. The weathering-enhancing effect of forestry is also seen in the first part of a forest rotation for silicate weathering, whereas an aging forest has slightly decreasing weathering rates. Increasing temperatures combined with the forestry induced weathering dynamic with higher weathering in young forest, produces a step-like increase in weathering rates of silicates in the BSC scenario (Figure 7).

## 4 Discussion

### 4.1 Implications of model differences

The weathering calculations in PROFILE and ForSAFE are based on the same equations, but in ForSAFE they are dynamic, while PROFILE has no time dimension. The models also differ in that several processes are only given as input data into PROFILE while they are modelled dynamically with ForSAFE and that feedbacks between these processes affect the system

in ForSAFE. In the PROFILE model, lack of nutrients because of low weathering can never affect tree growth, since uptake of nutrients to trees are input data. Low soil moisture during summers can also never affect weathering rates in PROFILE, because there are neither seasons nor modelled soil moisture values. PROFILE was developed at a time when climate change was usually not considered, to answer the question of what long term loads of acidity the ecosystem could tolerate (under the premise of unchanging forestry and climate), and for this it was sufficient. As acidification loads decreased, the role of forestry intensity for recovery from acidification increased (Iwald et al., 2013), A more complex model was needed and ForSAFE was developed, which include these processes and feedbacks. We have shown that despite their differences, the two models produce comparable estimates of weathering rates on these two sites.

The PROFILE model has often been used for critical load assessments and weathering estimates. This study shows that the more advanced model ForSAFE is as reliable as the PROFILE model and can be used to gain more information on the variation in weathering rates due to forestry practices, climate changes and temperature change, which could increase our understanding of the dynamics of ecosystem sensitivity. General conclusions regarding acid sensitivity, critical loads and the sustainability of forestry would not change significantly, but our ability to make customised or more detailed forestry plans with regards to intensity of harvest or to take acidification countermeasures would be improved.

## 4.2 Weathering dynamics in a changing environment

Another parameter that has a significant influence on weathering rates is the temperature. The climate is becoming warmer, and in some regions in Sweden, as elsewhere, it is possibly also becoming drier in the summer (Kjellström et al., 2018). Higher temperatures increase weathering, as shown in our simulations. However, drier conditions inhibit weathering, and dry periods in the summer, when weathering otherwise would be much higher than in the rest of the year, might affect the yearly weathering considerably. These two sites, although having lower soil moisture in the summer on average (Figure 2), does not seem to experience really dry summers more often in these future scenarios than during the 20th century, for the same forest stand age. Future studies, on regions that are believed to become much drier in summer in the future may help elucidate this. Akselsson et al. (2016) calculated the increase in weathering rate due to climate change in the 21st century in Sweden, using the PROFILE model. They found that the increase in weathering rates due to temperature increase up to 2050 varied at different locations in Sweden. The median increase in base cation weathering rate was 20 % for the ECHAM projection and 33 % for the HADLEY projection, which are both equivalent to about 10 % $°C^{-1}$. This is slightly higher than our result of a 7 % increase per degree increase in temperature. The difference is due to the fact that ForSAFE is a more complex model, with dynamic feedbacks between the uptake by trees, soil solution chemistry, soil moisture and weathering.

Forestry also affects weathering. After clear cutting, both soil moisture and soil temperature increase, leading to an increase in weathering rate. As uptake of nutrients to trees are halted and as the remaining litter starts to decompose, concentrations of base cations start to increase (Piirainen et al. 2004). Base cations in soil solution inhibit weathering of base cations in the model (like inorganic aluminium inhibit weathering of aluminium), but the increase in base cations is not sufficient to reduce the rate of weathering, since the soil moisture is still high. With whole-tree harvesting, much of the litter is removed, so that

there are less base cations to be released to soil water through decomposing, and the concentrations of base cations should not increase as much as with stem only harvesting (Ågren et al. 2010). This might be the reason for the very slight increase in weathering following whole-tree harvesting compared to stem only harvesting, found in this study. If base cation concentrations do not increase as much after whole-tree harvesting as after stem only harvesting, this also leads to less

leaching of base cations after whole-tree harvest than after stem only harvest. The slightly increased weathering rate and the decreased leaching may explain the diminishing difference in soil conditions with time between whole-tree harvesting and stem harvesting that has been seen in field experiments, despite the fact that a large quantity of base cations is removed from the ecosystem by whole-tree harvesting (Zetterberg et al., 2013).

According to ForSAFE, the weathering of silicate minerals is considerably suppressed by the atmospheric deposition of

acidifying substances, whereas the weathering of apatite (P and some of the Ca) was enhanced. The reason for this is the combined effects of $H^+$ as a driver of weathering and $Al^{3+}$ as an inhibitor of silicate weathering, but not of apatite weathering, since apatite does not contain Al. The solubility of Al increases with lower pH, thus inhibiting the weathering of silicates as the soil acidifies.

### 4.3 Model limitations and development

The results of this study demonstrate the importance of soil moisture on weathering rates. In PROFILE the soil moisture is an input, previously known from uncertainty studies to be of great importance for the weathering rates (Jönsson et al. 1995, Barkman and Alveteg, 2001), but often based on observation of the site and rough assumptions, whereas it is modelled in ForSAFE with soil texture, precipitation and temperature as inputs. For these two sites, average soil moisture modelled by ForSAFE is similar to the rough estimates of moisture used as input for PROFILE for most of the soil layers. The soil

moisture modelled by ForSAFE is also close to the calculated field capacity most of the time. Average soil moisture being close to field capacity could partly be an effect of the monthly time step, which evens out precipitation and gives enough time for draining of excess water each time step. A new version of ForSAFE with a daily time step is under development. A daily time step, with a more realistic time distribution of precipitation, with rainfall events and dry periods in between, affects the calculations of soil moisture on the short term, might affect the seasonal average soil moisture values and might

thus affect the predicted weathering rates; giving a greater variability in weathering between drier and wetter periods and potentially shifting the average. A shorter time step would potentially give more accurate results, given that soil moisture is an important parameter for weathering and soil moisture is highly variable on a shorter time scale than monthly.

In the SWETHRO sites, soil moisture and soil temperature are not measured and thus modelled soil moisture can't be compared to measured values. Another future study could model sites were such measurements are made and compare these,

for the weathering important parameters, with measurements.

Both the PROFILE model and the ForSAFE model are known to overestimate weathering in the lower soil layers (Stendahl et al., 2013; Zanchi, 2016). The soil horizon C consists of the less weathered parent material at the bottom of the soil profile, where weathering rates are low because the conditions in the soil inhibits weathering, despite the relative abundance of

weatherable minerals. Both PROFILE and ForSAFE currently calculate rather high weathering rates in the C horizon, if this soil layer is included in the calculations. In the modelling presented in this paper only a few centimetres of the C horizon are included, thus the total contribution of weathering from horizon C is small, but the rates per soil volume are equivalent to the layers above. Most of the C-horizon is usually located below the root zone, usually defined as the uppermost 50 cm of
mineral soil for spruce forests in Sweden, where more than 90 % of the spruce roots can be located (Rosengren and Stjernquist, 2004) and therefore not included in the modelling. The overestimation of weathering in the lower soil layers by these two models is likely to be, at least partly, due to the lack of calculation of the concentrations of dissolved silica in the soil water in both models. The dissolved silica, being a product of weathering of silicate minerals, acts as an inhibitor on the weathering of these minerals, i.e. all the minerals modelled in this study except apatite. The concentration of dissolved silica
in the soil water is currently being included in the ForSAFE model.

When the PROFILE and ForSAFE weathering profiles at Västra Torup and Hissmossa are compared to weathering rates at a nearby site, Skånes Värsjö, calculated with the depletion method (Stendahl et al., 2013), PROFILE and ForSAFE predict substantially higher weathering rates in the lower soil horizons, in line with the above discussion on overestimation in the lower layers. The weathering rates modelled in the upper horizons by PROFILE and ForSAFE are, on the other hand, lower
than the rates obtained with the depletion method. However, the depletion method does not calculate present-day weathering, but average weathering in the soil layer since deglaciation. The weathering rates have varied with time, both because new soils have more easily weatherable material and weathers much faster than older soils (Starr and Lindroos, 2006) and because environmental conditions have varied since the end of the last glaciation. This means that weathering rates calculated with methods that calculate average weathering since the deglaciation, such as the depletion method, should
generally be higher than PROFILE and ForSAFE weathering rates, except for the lower soil layers, since the weathering front moves down .

## 5 Conclusions

We have shown that despite the differences between PROFILE and ForSAFE, the two models give comparable estimates of annual weathering rates.
The PROFILE model has often been used for critical load assessments and weathering estimates. This study shows that the more advanced model, ForSAFE, can be used to gain much more information on the variation in weathering rates in response to forestry and climate change.

The results from ForSAFE presented in this paper demonstrate that weathering rates vary considerably; between seasons, between years and on longer time scales. This dynamic behaviour can be of importance in nutrient leaching and nutrient
availability to the trees: during seasons with high nutrient demand there might be risk of nutrient deficiency, even though there might be higher availability of nutrients than demand and nutrient losses through leaching during other seasons.

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

**Table 1. Characteristics of the two sites.**

| | Västra Torup | Hissmossa |
|---|---|---|
| Coordinates (°N, °E) | 56.135, 13.510 | 56.181, 13.515 |
| Active years | 1988 - 2010 | 2010 - |
| Year of planting | 1941 | 1973 |
| Year of clear cutting | 2010 | - |
| Standing stem biomass (g m$^{-2}$) (year in parenthesis) | 18841 (2010) | 10559 (2011) |
| **Measured throughfall** | | |
| Precipitation (mm) | 430 - 780 | 460 - 730 |
| S deposition (kg ha$^{-1}$ y$^{-1}$) | 4.5 - 27 * | 3.6 - 6.9 |
| N deposition (kg ha$^{-1}$ y$^{-1}$) | 6.2 - 12 | 6.8 - 11 |
| Cl deposition (kg ha$^{-1}$ y$^{-1}$) | 21 - 50 | 33 - 87 |
| Ca+Mg+Na+K deposition (kg ha$^{-1}$ y$^{-1}$) | 31 - 57 | 39 - 80 |
| **Measured soil water chemistry** | | |
| pH | 4.4 - 4.9 | 4.2 - 4.5 |
| SO$_4$-S (mg l$^{-1}$) | 0.8 - 7.3 | 2.1 - 4.4 |
| Cl (mg l$^{-1}$) | 3.2 - 20 | 17 - 51 |
| NO$_3$-N (mg l$^{-1}$) | 0 - 0.1 | 0.5 - 3.3 |
| NH$_4$-N (mg l$^{-1}$) | 0 - 0.2 | 0 - 0.1 |
| Ca (mg l$^{-1}$) | 0.2 - 1.0 | 0.2 - 1.7 |
| Mg (mg l$^{-1}$) | 0.2 - 1.0 | 0.6 - 1.9 |
| Na (mg l$^{-1}$) | 2.8 - 8.4 | 12 - 23 |
| K (mg l$^{-1}$) | 0.1 - 1.1 | 0.2 - 1.0 |
| Inorganic Al (mg l$^{-1}$) | 0.2 - 3.4 | 0.6 - 5.3 |
| Organic Al (mg l$^{-1}$) | 0 - 0.4 | 0.6 - 1.1 |
| Al-tot (mg l$^{-1}$) | 0.4 - 3.7 | 1.5 - 6.2 |
| TOC (mg l$^{-1}$) | 3.5 - 15 | 8.2 - 21 |

* Decreasing steeply with time

**Table 2. Measured soil parameters for the five soil layers (O, A, AB, B and C) at the two sites. Above: thickness of layer, bulk density, percentage organic matter, estimated percentage stones, measured size fractions, pH, exchangeable ions, cation exchange capacity, base saturation, fraction of carbon and nitrogen. Below: total chemistry of all dry soil matter.**

| Horizon | Thickness | Bulk density | OM | Stoniness | Clay | Silt | Sand | pH $H_2O$ | Al | H | Na | K | Mg | Ca | CEC | BS | Tot-C | Tot-N |
|---|---|---|---|---|---|---|---|---|---|---|---|---|---|---|---|---|---|---|
| | (m) | (kg m$^{-3}$) | (% of DW) | (%) | (% of mineral soil) | | | | Exchangeable ions ($\mu$eq g$^{-1}$) | | | | | | ($\mu$eq g$^{-1}$) | (%) | (g (kg DW)$^{-1}$) | |
| **Västra Torup** | | | | | | | | | | | | | | | | | | |
| O | 0.05 | 181 | 87 | 0 | | | | 4.0 | 29 | 84.5 | <0.1 | 13.0 | 27.9 | 50.1 | 205 | 43.0 | 543 | 20.9 |
| A | 0.06 | 959 | 6 | 20 | 5 | 27 | 68 | 4.1 | 31 | 16.5 | <0.1 | 1.0 | 0.7 | 0.8 | 50 | 4.9 | 34 | 2.0 |
| AB | 0.20 | 1062 | 5 | 20 | 5 | 31 | 64 | 4.6 | 27 | 6.1 | <0.1 | 0.6 | 0.4 | 1.2 | 36 | 6.4 | 25 | 1.7 |
| B | 0.20 | 1279 | 4 | 20 | 3 | 21 | 76 | 4.8 | 16 | 1.3 | <0.1 | 0.4 | 0.1 | 0.6 | 18 | 6.5 | 18 | 1.3 |
| C | 0.04 | 1446 | 2 | 20 | 0 | 17 | 83 | 4.9 | 13 | 4.8 | <0.1 | 0.4 | 0.1 | 0.5 | 19 | 5.0 | 8 | 0.6 |
| **Hissmossa** | | | | | | | | | | | | | | | | | | |
| O | 0.05 | 394 | 65 | 0 | | | | 3.5 | 45 | 63.5 | 3.7 | 8.1 | 21.4 | 26.4 | 164 | 36.8 | 391 | 19.8 |
| A | 0.13 | 909 | 8 | 10 | 0 | 5 | 91 | 3.8 | 36 | 15.3 | 0.6 | 1.4 | 3.8 | 2.9 | 60 | 17.6 | 46 | 3.0 |
| AB | 0.10 | 1075 | 8 | 10 | 1 | 8 | 89 | 4.6 | 27 | 4.6 | 0.4 | 0.8 | 2.5 | 2.3 | 37 | 17.8 | 38 | 3.0 |
| B | 0.28 | 1276 | 3 | 10 | 0 | 9 | 88 | 4.5 | 12 | 0.3 | 0.4 | 0.7 | 2.3 | 2.3 | 17 | 34.2 | 17 | 2.2 |
| C | 0.04 | 1316 | 3 | 10 | 0 | 8 | 88 | 4.7 | 11 | 0.7 | 0.4 | 0.7 | 2.4 | 2.3 | 17 | 36.1 | 14 | 2.0 |

| Horizon | Si | Al | Ca | Fe | K | Mg | Mn | Na | P | Ti |
|---|---|---|---|---|---|---|---|---|---|---|
| | Total chemistry of mineral and organic matter (mg (kg DW)$^{-1}$) | | | | | | | | | |
| **Västra Torup** | | | | | | | | | | |
| O | 33100 | 5060 | 1760 | 2540 | 2540 | 623 | 167 | 1440 | 789 | 337 |
| A | 349000 | 50000 | 7110 | 20000 | 26300 | 1540 | 386 | 15900 | 253 | 4280 |
| AB | 335000 | 56400 | 8190 | 28600 | 26600 | 2610 | 469 | 16300 | 401 | 4510 |
| B | 341000 | 58600 | 8920 | 22600 | 27700 | 2950 | 436 | 17100 | 540 | 3700 |
| C | 348000 | 59500 | 9820 | 24400 | 28700 | 3240 | 493 | 17800 | 602 | 4240 |
| **Hissmossa** | | | | | | | | | | |
| O | 77300 | 13500 | 1970 | 4370 | 5710 | 580 | 103 | 3540 | 787 | 844 |
| A | 322000 | 51500 | 5040 | 19200 | 28400 | 822 | 277 | 14600 | 166 | 3540 |
| AB | 322000 | 65700 | 7030 | 29100 | 29600 | 2070 | 459 | 16900 | 217 | 3340 |
| B | 340000 | 66000 | 7490 | 27600 | 30700 | 2140 | 419 | 18900 | 231 | 3290 |
| C | 329000 | 70500 | 8840 | 28800 | 33600 | 2670 | 822 | 19000 | 451 | 3030 |

**Table 3. Soil input data to the models, standard values (partial pressure of $CO_2$ and gibbsite constant) or calculated from measured soil parameters at the two sites (mineral area, field capacity, wilting point, field saturation and percentage of minerals). The modelled layers L1 - L5 correspond to soil layers O, A, AB, B and C in the two soils. Hissmossa L5 is below the modelled root zone of 50 cm.**

| Layer | Mineral area | pCO$_2$ | Kgibb | FC | WP | FS | Quartz | K-feldspar | Albite | Anorthite | Muscovite | Epidote | Hornblende | Apatite | Illite | Vermiculite1 | Vermiculite2 | Chlorite1 | Chlorite2 |
|---|---|---|---|---|---|---|---|---|---|---|---|---|---|---|---|---|---|---|---|
| | (10$^6$ m$^2$ m$^{-3}$) | | | (m$^3$ m$^{-3}$) | | | SiO$_2$ | KAlSi$_3$O$_8$ | NaAlSi$_3$O$_8$ | CaAl$_2$Si$_2$O$_8$ | a | b | c | d | e | f | g | h | i |
| **Västra Torup** | | | | | | | | | | | | | | | | | | | |
| L1 | 214161 | 10 | 6.5 | 0.31 | 0.11 | 0.87 | 50 | 17 | 19 | 2.2 | 2.4 | 2.0 | 0.4 | 0.2 | 1.3 | 0.6 | 0.2 | 0.4 | 0.2 |
| L2 | 1131959 | 20 | 7.6 | 0.21 | 0.06 | 0.68 | 50 | 17 | 19 | 2.2 | 2.4 | 2.0 | 0.4 | 0.2 | 1.3 | 0.6 | 0.2 | 0.4 | 0.2 |
| L3 | 1334007 | 20 | 8.6 | 0.24 | 0.06 | 0.64 | 46 | 16 | 19 | 2.4 | 4.4 | 2.1 | 0.7 | 0.2 | 2.3 | 1.0 | 0.4 | 0.6 | 0.4 |
| L4 | 1167398 | 20 | 9.2 | 0.22 | 0.06 | 0.54 | 45 | 16 | 20 | 2.5 | 4.1 | 2.2 | 0.8 | 0.3 | 2.2 | 1.1 | 0.5 | 0.7 | 0.5 |
| L5 | 909226 | 20 | 9.2 | 0.18 | 0.03 | 0.47 | 44 | 17 | 20 | 2.6 | 3.3 | 2.3 | 0.9 | 0.3 | 1.9 | 1.3 | 0.5 | 0.8 | 0.5 |
| **Hissmossa** | | | | | | | | | | | | | | | | | | | |
| L1 | 143372 | 10 | 6.5 | 0.42 | 0.17 | 0.80 | 42 | 18 | 17 | 1.5 | 1.2 | 1.4 | 0.2 | 0.1 | 2.9 | 0.3 | 0.1 | 0.2 | 0.1 |
| L2 | 330775 | 20 | 7.6 | 0.18 | 0.05 | 0.65 | 42 | 18 | 17 | 1.5 | 1.2 | 1.4 | 0.2 | 0.1 | 2.9 | 0.3 | 0.1 | 0.2 | 0.1 |
| L3 | 491284 | 20 | 8.6 | 0.20 | 0.06 | 0.58 | 39 | 17 | 19 | 2.1 | 2.9 | 1.8 | 0.6 | 0.1 | 5.2 | 0.7 | 0.3 | 0.4 | 0.3 |
| L4 | 534872 | 20 | 9.2 | 0.14 | 0.03 | 0.51 | 41 | 17 | 21 | 2.5 | 4.8 | 1.7 | 0.5 | 0.1 | 3.5 | 0.7 | 0.3 | 0.4 | 0.3 |
| L5 | 538935 | 20 | 9.2 | 0.15 | 0.03 | 0.50 | 37 | 18 | 21 | 2.7 | 5.5 | 2.0 | 0.7 | 0.2 | 3.7 | 0.9 | 0.4 | 0.5 | 0.4 |

a  $K_{44}Na_2Mg_8Fe_{12}Ti_2Al_{96}Si_{120}O_{390}(OH)_{94}$
b  $Ca_{80}Fe_{30}Al_{96}Si_{124}O_{495}(OH)_{44}$
c  $K_{18}Na_{54}Ca_{166}Mg_{210}Fe_{180}Ti_{11}Al_{216}Si_{606}O_{2146}(OH)_{188}$
d  $Ca_{10}(PO_4)_6(OH)_2$
e  $K_{0.6}Al_2(Al_{0.6}Si_{3.4}O_{10})(OH)_2$
f  $Ca_{20}Mg_{103}Fe_{182}Al_{162}Si_{293}O_{832}(OH)_{804}$
g  $Ca_{10}Mg_{103}Fe_{22}Al_{68}Si_{123}O_{249}(OH)_{490}$
h  $Na_2Ca_3Mg_{107}Fe_{124}TiAl_{124}Si_{138}O_{540}(OH)_{442}$
i  $Mg_{103}Fe_{58}TiAl_{100}Si_{87}O_{365}(OH)_{302}$

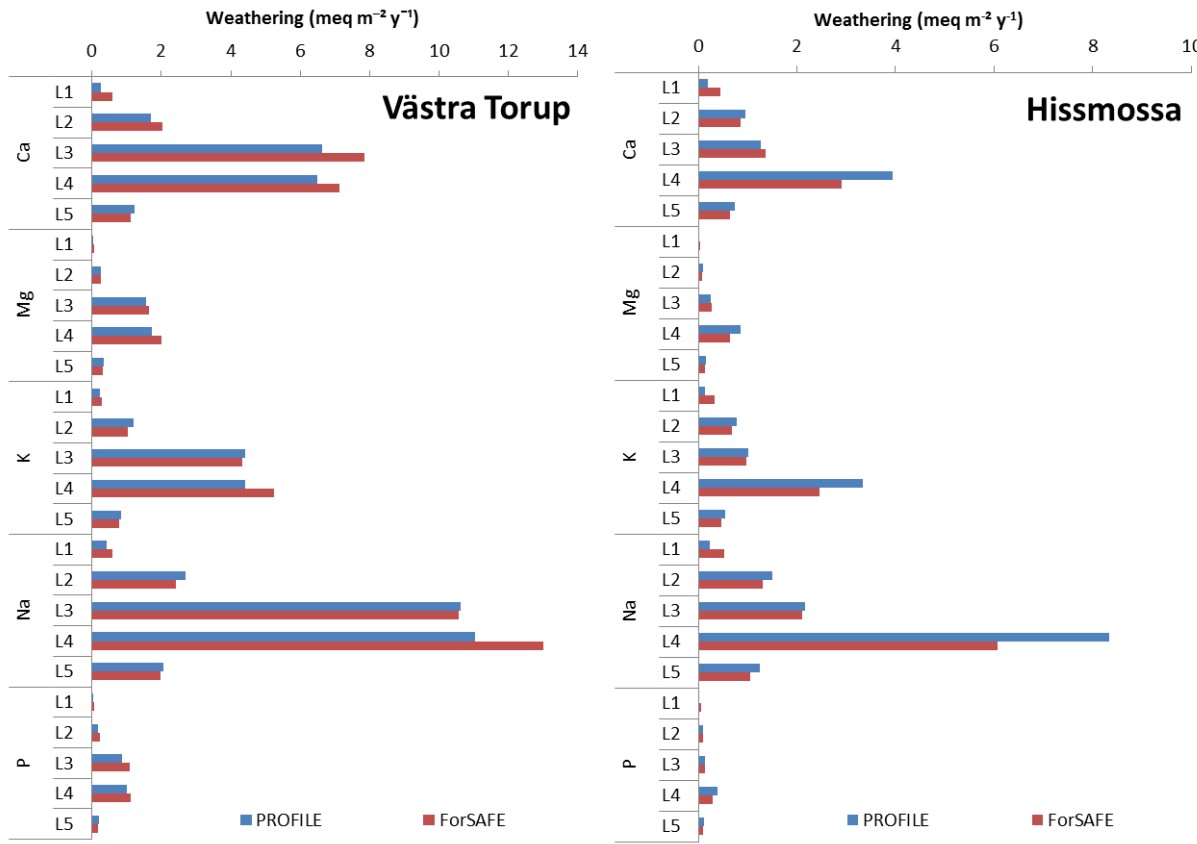

**Figure 1. Weathering rates (meq m$^{-2}$ y$^{-1}$) calculated with the PROFILE model and the ForSAFE model (averages over a forest rotation, BSC scenario), for the sites at Västra Torup and Hissmossa, for soil layers L1 (top layer) to L5 (bottom layer at ~50 cm depth). The time period is from one clear cut to the next and is different for the two sites: 2011-2080 for Västra Torup and 2041-2100 for Hissmossa. Note that the rates are shown here per layer, so that the bars show directly how much of the total weathering each soil layer contributes. For Hissmossa. L5 is shown, even though it lies below the root zone and is not included in calculations of weathering rates in the root zone.**

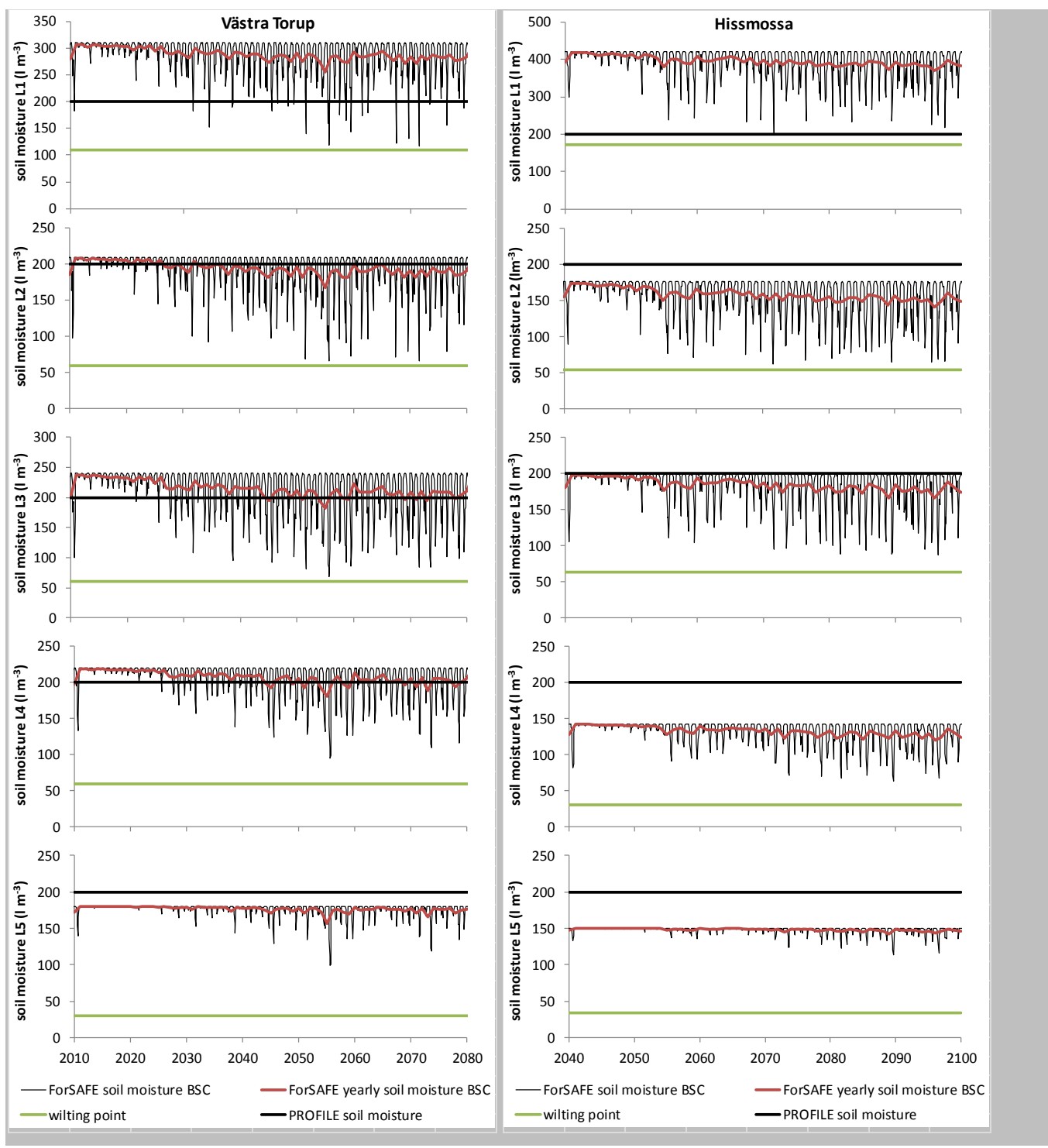

**Figure 2. Soil moisture in all soil layers, BSC scenario, forest rotation 2010-2080 in Västra Torup and 2040-2100 in Hissmossa, compared to wilting point and PROFILE input soil moisture.**

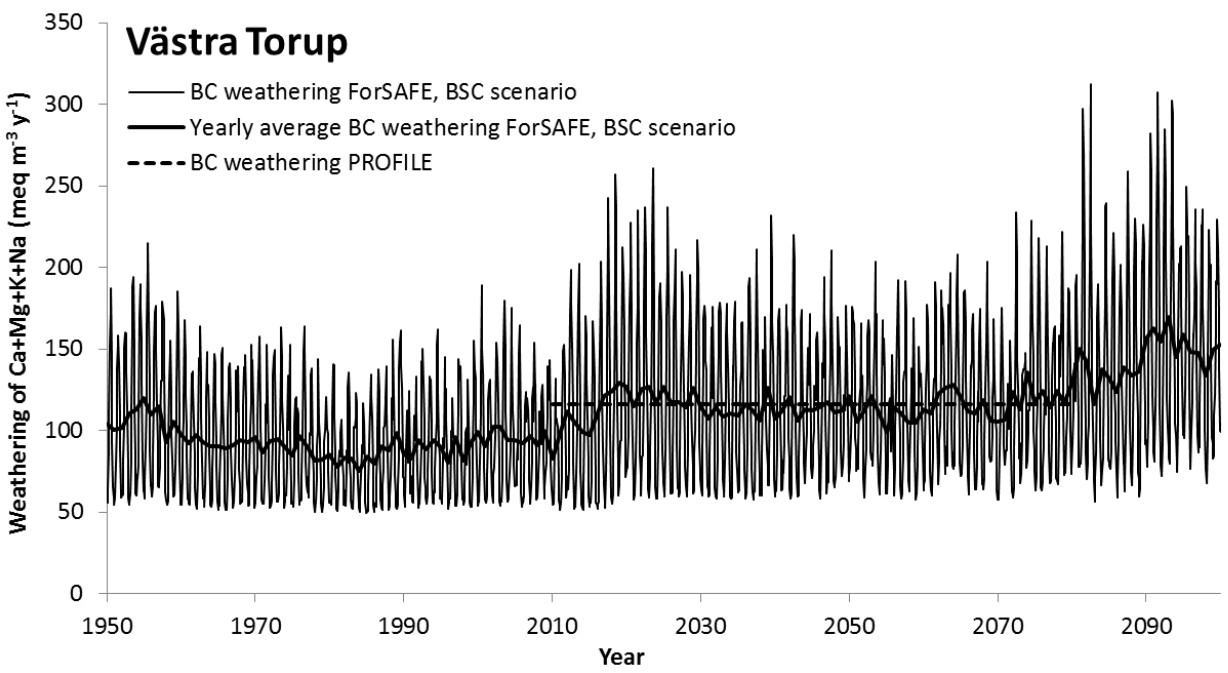

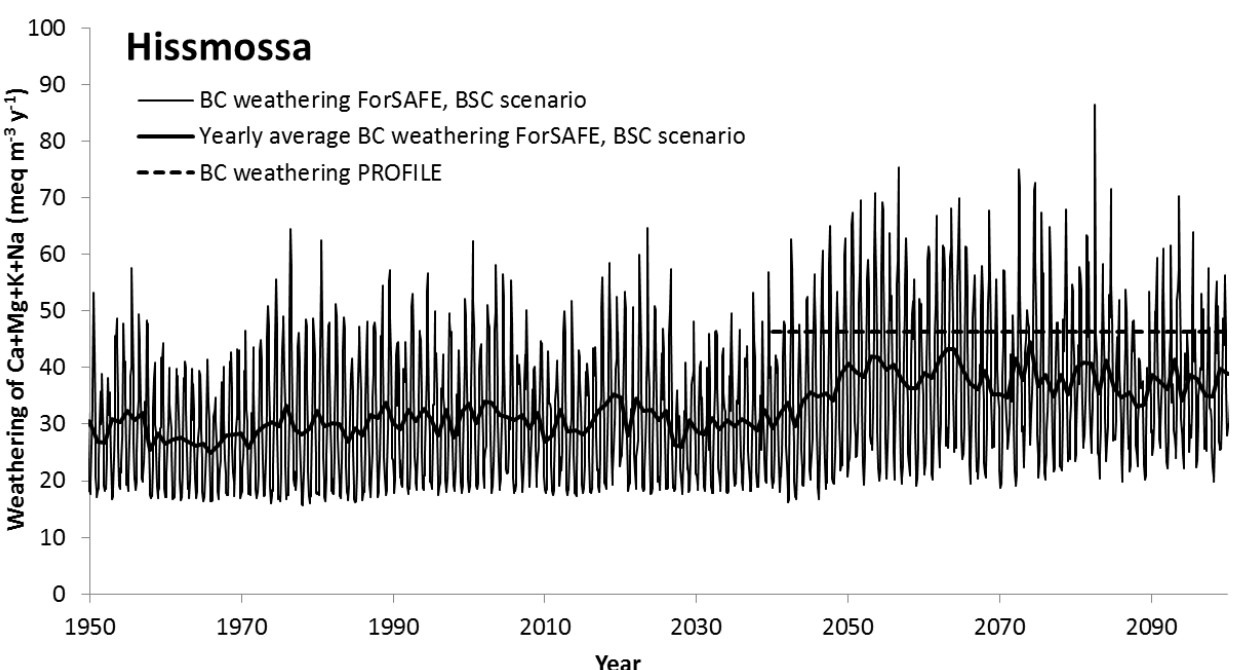

Figure 3. Modelled Ca+Mg+K+Na weathering in Västra Torup (above) and Hissmossa (below) from 1950 to 2100 (note the difference in scale for the two sites). PROFILE calculates the average weathering rates for the time period represented by the input values, while monthly weathering values were calculated with ForSAFE, using the BSC scenario.

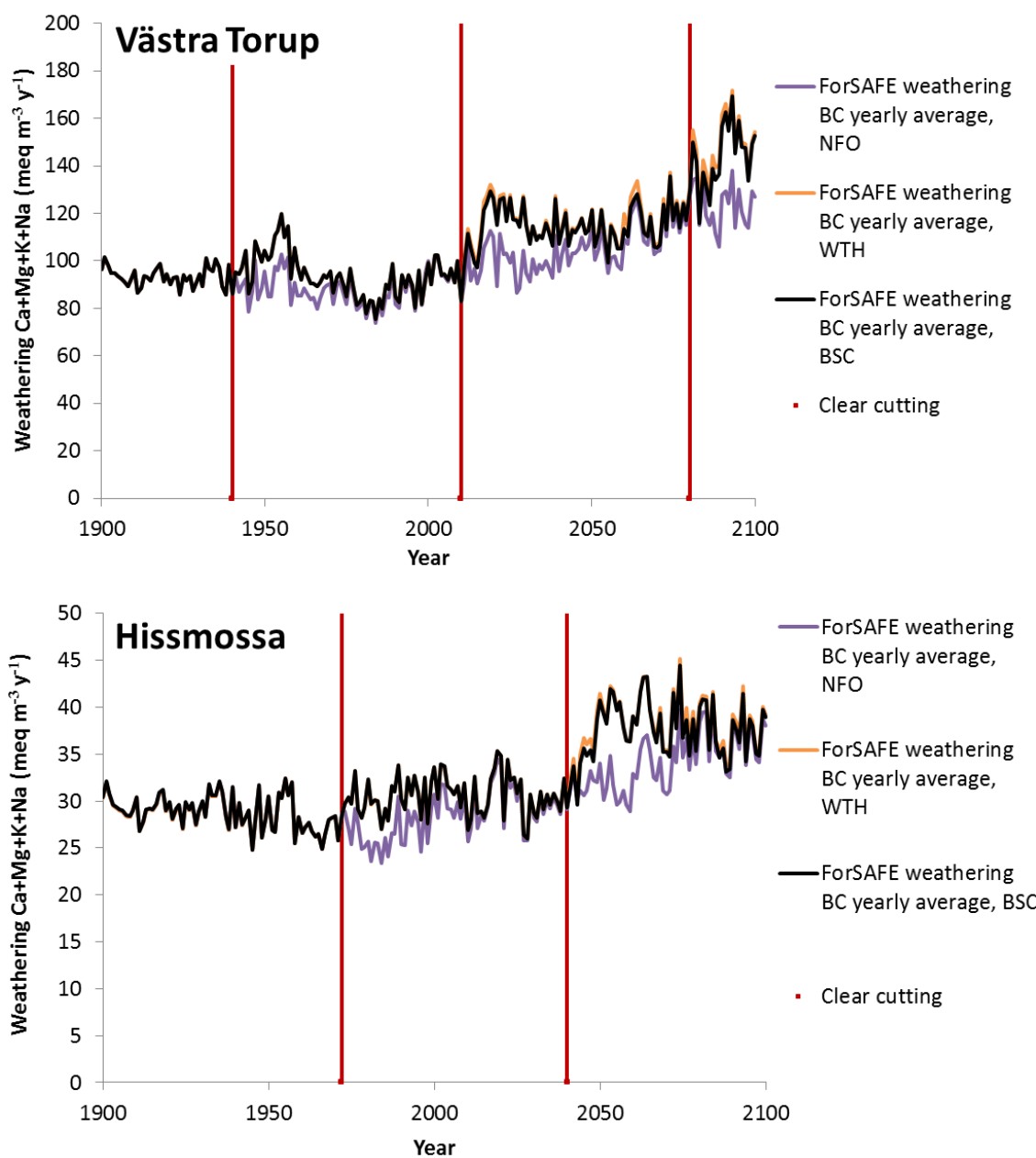

**Figure 4. Yearly average weathering of base cations in the whole soil profile, for the BSC scenario, the whole-tree harvest WTH scenario, and the NFO scenario without any clear cutting or thinning. The years of clear cuts in the BSC and WTH scenarios are marked with vertical lines. In Västra Torup clear cuts are in the years 1940, 2010 and 2080 and in Hissmossa in 1972, 2040 and 2101.**

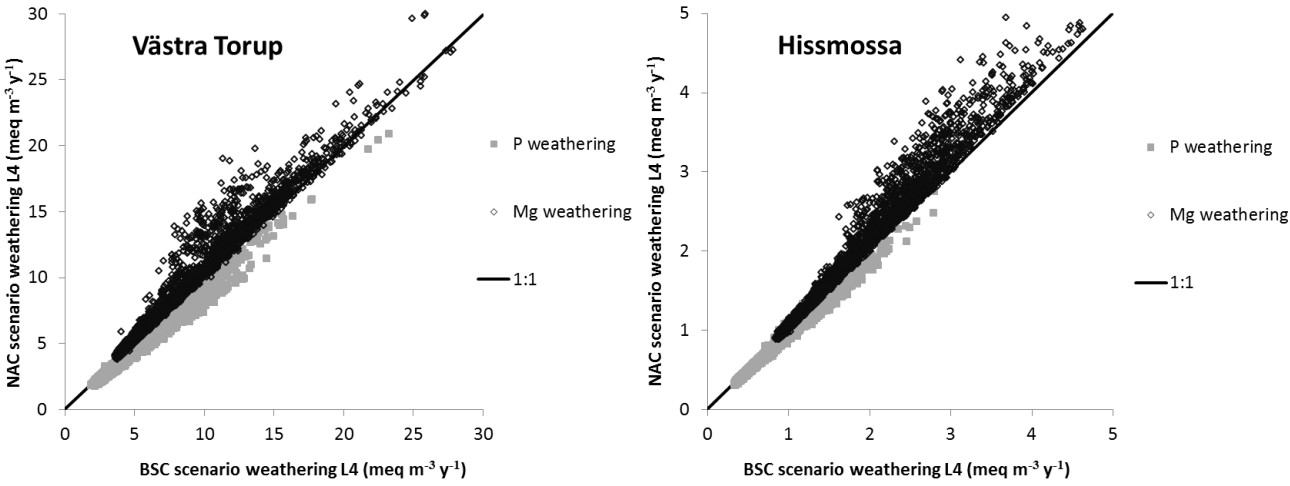

**Figure 5. Comparison of weathering rates of Mg and P in soil layer L4 in the non-acidification scenario NAC and the base scenario BSC (meq m$^{-3}$ y$^{-1}$). With the mineralogy of these sites, Mg is only weathered from silicate minerals and P is only weathered from apatite. One dot represent one month.**

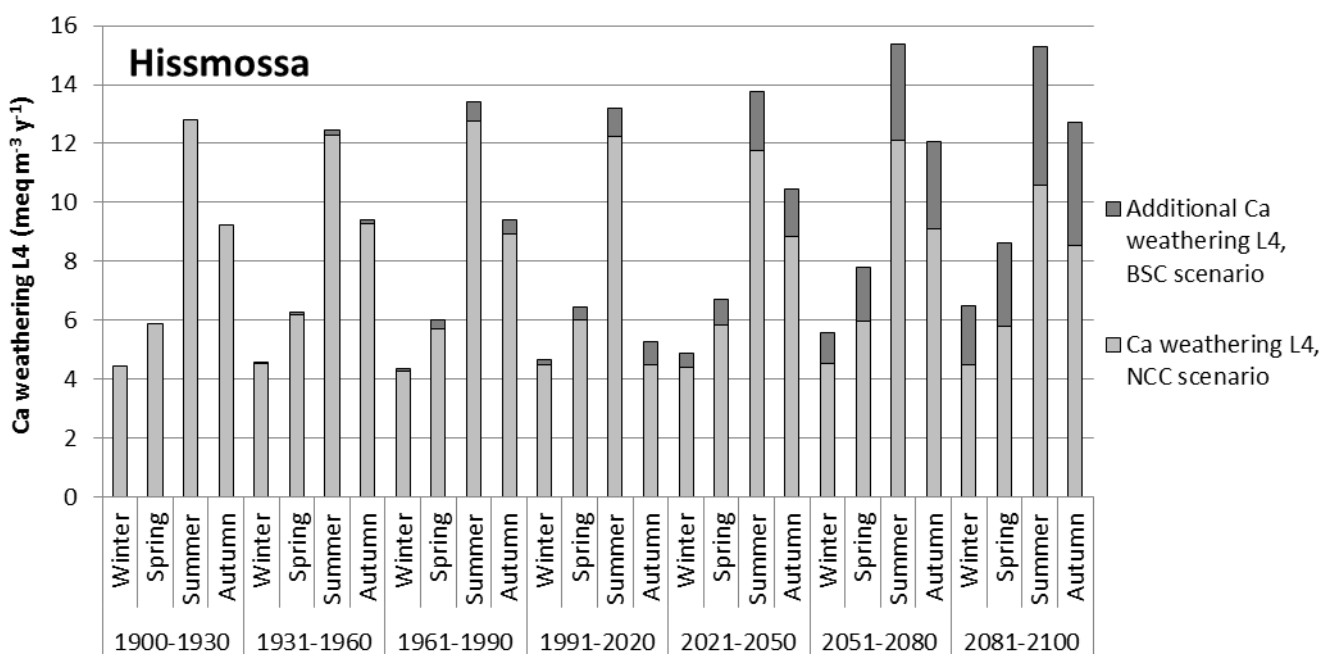

**Figure 6. The effect of the increased temperature of the BSC scenario on Ca weathering in L4 at Hissmossa, compared to the NCC scenario with no climate change, shown as averages for seasons over periods of 30 years. Winter = December, January and February, spring = March, April and May, summer = June, July and August and autumn = September, October and November.**

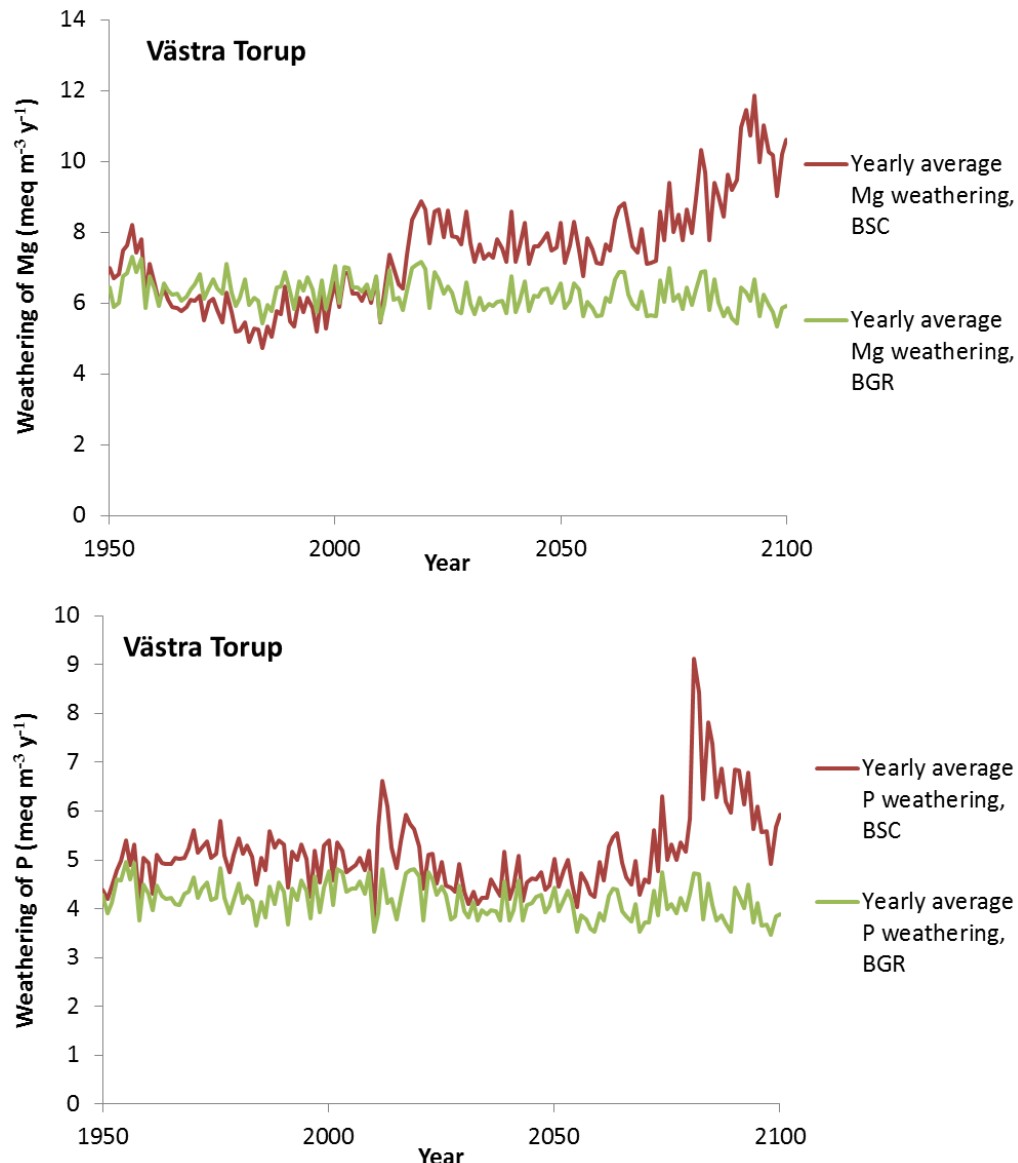

**Figure 7. Weathering of Mg (from silicates) and P (from apatite) at Västra Torup, under the BSC scenario and the BGR scenario with neither acidification, climate change nor forestry.**

