# Peer review of "Dynamic modelling of weathering rates – the benefit over steady-state modelling"

_SOIL, 2018_

## Referee Comment (RC1) · Anonymous Referee #1 · 13 Aug 2018

A neat and generally well written and structured paper, the subject of which (weathering) falls within the scope of SOIL and is of broad international interest. However, several related and somewhat similar applications of the models have already been published, as indicated by the strong Sweden related references. And - as stated by the authors - as both models are based on the same weathering equations, it is hardly surprising that the results (long-term) from both models are similar and suggests little benefit is gained in using the more complex model when looking at long-term climate change and forest management impacts. Furthermore, the dynamic ForSAFE model, by definition is bound to provide more detailed and seasonal results and information, making the answers to the question posed by the title and to the hypotheses somewhat self-evident. I think the paper could be substantially improved if there were more focus on and calibration of the ForSAFE model with empirical field data; for example, with measured soil temperature, soil moisture, forest growth (base cation uptake) and leaching data.

Specific comments relating to validity of analyses and assumptions, relevance of discussion and conclusions.

p. 1, l. 20 (and p. 4, l. 31). Annual precipitation may remain similar but what about the seasonal distribution? How will temperature change affect snowfall and snowmelt, surely a very important feature of climate change in such latitudes, the water cycle and weathering?

p. 3, l. 19 (and p. 8, l.11). Why/how are base cations, Al3+ and organic acids inhibitors of weathering? Do you mean that if the concentrations of weathering products in the soil solution increase, the weathering reaction slows down as equilibrium concentrations are reached? But then aren't the weathering products being continually take away through up take, leaching or adsorption by the soil allowing weathering to proceed?

p. 4, l.16. There is little description of the soil type at the two sites. From the horizon abbreviations listed in Table 2, it would appear the soil are not Podzols?

p. 5, l.16. Furthermore, thickness of the mineral soil horizons at Hissmossa is 55 cm (and not 50 cm; Table 2) and how/why is the organic layer included where surely it is a question of organic matter decomposition rather than mineral weathering? It is not stated how stoniness was derived and if the hydraulic parameter values (Table 3) have been corrected for stone content. Given the concluded importance of soil moisture, the fixed value used in PROFILE and same value for all layers (0.2) seems rather crude. The field capacity values in Table 3 also seem somewhat low – is this because of stone content correction? How have any time related changes in organic layer thickness (and therefore soil moisture content) been taken into account? P. 4, l. 31: by "rainfall" you mean annual precipitation?

p. 5, l. 6-11. The description of the scenarios is unclear, at least to me. For example, does the base scenario mean there are two thinnings and a clear cut every 70 years (and starting from the year of planting – Table 1), deposition loads constant from "todays" (which year?) levels into the future plus climate change temperature (but no change in precipitation)? What is the whole tree harvesting treatment: stems + branches or stems + branches + stumps? Is it carried out every 70 years during the 1900-2100 period? It would be useful to number or letter the scenarios and refer to them in the text and, tables and figures. Which scenario is used for Figure 1? Doesn't the 70 year rotation period cover a different set of years between the two sites (2011-2080 vs. 2041-2100), when the climate change has changed the climate. Doesn't this explain the differences in weathering between the two sites rather than differences in soil texture (p. 5, l. 21), which anyway would also affect the soil hydraulic properties besides surface area? Why is only Mg weathering presented in Figure 2 and in Figure 4 to represent silicate mineral weathering? Wouldn't the sum of base cations be a more appropriate measure of overall silicate weathering? As weathering largely takes place by acid (proton) attack, why is silicate weathering decreased by acidified conditions (p. 6, l. 17) and why would apatite weathering be increased? Doesn't the base scenario include harvesting effects besides climate change effects (p. 6, l. 29)? See also p. 8, l. 21-24.

p. 7. l. 19: by "more detailed forestry plans" do you mean timing of thinning and timing and intensity of harvesting?

p. 8, l. 13- Isn't a matter of litter decomposition and not weathering? And I think you need to give a reference that supports the statement about harvesting intensity effects of soil solution base cation concentrations. Concentrations and leaching loads may increase with whole-tree harvesting as a result of increased drainage (percolation) and disturbance of the site.

p. 8. L. 25-. The contribution of your C-horizon is small (Table 2). And if the material is less weathered, then there would be more weatherable minerals and therefore potential

for weathering? Is the explanation for taking silica concentrations into account the same as mentioned above, i.e. equilibrium concentrations reached?

p. 9. l. 6. A paper by Starr & Lindroos (Geoderma (2006) 133: 269–280) shows this.

Conclusions: I appreciate the recognition of the importance of soil moisture to weathering (and decomposition) and for the reasons stated (time step). However, wouldn't a model with a daily time step be more suited to forest stand nutrient/biogeochemical cycling studies rather than modelling long-term climate change effects.

Units: Wouldn't it be more correct to present weathering in units of moles charge rather than equivalents? And why sometimes meq m-2 yr-1 and sometimes meq m-3 y-1 (Fig. 2, 3, 4, 5). Is it somehow because the latter refer to a specific layer (of differing thicknesses) rather than to the fixed organic layer + 50 cm layer or simply a typo?

Table 1. Coordinates are decimal degree latitude (N) and longitude (E). Figure 1. Title: Using which scenario and for which period of time? Figures 2 & 6. Is it necessary to include the ForSAFE monthly values? See Fig.3. Fig. 3. Use a circle around the years of clear cutting on the whole tree harvesting scenario and base scenario lines rather than the vertical line that intersects all scenarios. Are the years 1941,2010 and 2080 for Västra Torup and 1973 and 2043 for Hissmossa? Fig. 4. Is each dot is a year of the 70 year rotation period? Why only layer 4 (B-horizon)? Fig. 5. The base scenario includes the climate change temperature scenario and does the "constant climate change scenario" refer to the last one in the list on p. 5?

[Figure]

---

## Referee Comment (RC2) · Anonymous Referee #2 · 21 Sep 2018

The study presents soil weathering rates for two sites in Sweden. It is concise and well written, and the subject is within the scope of SOIL. However, I would urge the authors to rethink the focus of the manuscript; the comparison of output from a weathering sub-model within a steady-state model to output from the same weathering sub-model in a dynamic model is somewhat trivial, and the results are unsurprising. Similarly, it is also unsurprising that a dynamic model provides more temporal data compared to a steady-state model. The current aims of the manuscript are trivial. There is merit in presenting the dynamics of weathering; the Title and presentation could be refocused to 'Dynamic modelling of weathering rates – the benefit over steady-state modelling'. Secondly, the discussion somewhat repeats the results, there are few references to

literature, and overall it feels more like a report.

I have listed additional points for the authors consideration by page (P) and line (L) number.

P1L10. Do not mention SWETHRO in the abstract. P1L20. The results / discussion of scenario's does not fit with the objectives or title of the manuscript. P1L23. This is not a result of this study. P2L23. Given the importance of weathering (Title / objectives), it is surprising that weathering is given limited attention in the introduction. P2L27. I recommend that you remove the first objective and expand (refocus) the second. P2L30. What is the objective of the scenarios? P3L2. What were the models applied to two forests? P3L3. Please provide more background. For the external reader SWETHRO has no meaning or context. P3L17. Does 'factors affecting' mean sensitive parameters? Can you site previous sensitivity studies? P4L14. There is more sand at Hissmossa (Table 2) but more less Quartz (Table 3). How is this? P4L20. It is surprising that a fixed value is used for soil moisture given (a) that soil moisture is an 'important' parameter (as noted by the authors), and that (b) the determinants of soil moisture (texture, bulk density and organic matter) are very different between both sites (as noted by the authors). P4L21. The authors need to provide better context (justification) for the scenarios. P5L5. The list of scenarios suggests a study objective different than that presented. P5L25. Given the importance of soil moisture, why are these data not shown (Figure or Table) P5 Figure 1. The difference between PRO-FILE and ForSAFE in L4 (and L5) at Hissmossa needs more quantitative explanation / support. It might be soil moisture but this is not clearly shown. P6 Figures 2 to 6. Many of the figures forces on magnesium or calcium but the sum of base cations is the focus of the text (primarily). P7L10. The discussion has notably few citations... it is a discussion? P9L15–L19. These are not surprising conclusions (and more-or-less were previously known). Table 3. It appears that surface area is estimated for Clay, Silt and Sand. How are areas for the O horizon estimated?

---

## Author Response (AR1)

**Authors response: Comments on the manuscript "Dynamic modelling of weathering rates – Is there any benefit over steady-state modelling?"**

Authors: Veronika Kronnäs, Cecilia Akselsson and Salim Belyazid

veronika.kronnas@nateko.lu.se

5  MS No.: soil-2018-25

2018-11-27

**Contents**

We would like to thank both reviewers so much for your comments. They were very helpful and we think that they have helped us improve the manuscript and make it more clear and understandable. Below we give our answers to the questions the reviewers had.

**Comments from reviewer 1**

**GENERAL COMMENTS**

**Comment:** "A neat and generally well written and structured paper, the subject of which (weathering) falls within the scope of SOIL and is of broad international interest. However, several related and somewhat similar applications of the models have already been published, as indicated by the strong Sweden related references."

*Answer: Thank you! It is true that other outputs from ForSAFE (such as soil water chemistry) have been analysed and evaluated in a number of papers, but the actual weathering rates themselves have not been compared with any other methods. This study shows that the weathering calculations in ForSAFE give weathering rates of the same size as the much more tested and evaluated model PROFILE.*

**Comment:** "And - as stated by the authors - as both models are based on the same weathering equations, it is hardly surprising that the results (long-term) from both models are similar and suggests little benefit is gained in using the more complex model when looking at long-term climate change and forest management impacts."

*Answer: Both models are based on the same weathering equations, but in ForSAFE the equations are dynamic, while they are not in PROFILE. There are processes in ForSAFE, which are important for the weathering and which are not modelled*

*at all in PROFILE – soil hydrology for example. Also, the timing of different processes relative to each other can be important for the weathering and affect the weathering rates; also the long term average rates. This cannot be captured in PROFILE. For example the timing of the relatively high uptake to vegetation in the beginning of the vegetation period vs the timing of the high weathering during warmer months, and high leaching during wetter months – these processes affect soil*

5    *water chemistry which in its turn affect weathering in the next time step. Also, PROFILE assumes steady state and cannot take into consideration how the system got there, if the changes were slow and gradual or abrupt. Reality is never in steady state, and climate in the future is less in steady state now than in a long time, which makes a dynamic model more important than in earlier calculations of what loads of acidity the ecosystem in climatic steady state could tolerate. We tried to describe this in the discussion, P7L26-P8L8. See also our answer to the second reviewer's second question.*

10    **Comment:** "Furthermore, the dynamic ForSAFE model, by definition is bound to provide more detailed and seasonal results and information, making the answers to the question posed by the title and to the hypotheses some-  what self-evident."

*Answer: We have changed the title to "Dynamic modelling of weathering rates – the benefit over steady-state modelling" (P1L2), as suggested by the second reviewer. We have also added a bit to the second objective (P2L31-32): "...scenarios, representing important ecological issues: acidification, climate change and nutrient removal through land use.", since a*

15    *large part of the paper describes the weathering response to environmental drivers (i.e. describes in detail some benefits that can be gained by dynamic modelling). By doing those changes, the title and the second objective fits better to what we actually do in the paper –describing the benefits and the dynamics in weathering. For example, we show that even though ForSAFE weathering rates vary a lot, averages over the same period represented in the PROFILE modelling, fall so close to PROFILE values, which gives credit to the ForSAFE weathering. We didn´t know this before we performed the study.*

20    **Comment:** "I think the paper could be substantially improved if there were more focus on and calibration of the ForSAFE model with empirical field data; for example, with measured soil temperature, soil moisture, forest growth (base cation uptake) and leaching data."

*Answer: ForSAFE is not calibrated with empirical field data except base saturation and soil carbon and nitrogen. Other data can be used for comparisons with model results. Unfortunately we don't have measurements of soil moisture or soil*

25    *temperature for these sites. Since soil moisture and temperature are very important parameters, a very good next step would be to compare ForSAFE modelled soil moisture and soil temperature to measured data, but that is outside the scope of this paper, as that would have to be on other sites. We include this suggestion in the discussion (P9L28-30) as a suggested future study. There are measured forest growth and soil water concentrations for these sites, but there are already papers comparing ForSAFE modelled values with these kinds of measurements (Yu et al in the reference list and others).*

**SPECIFIC COMMENTS**

**Comment:** "p. 1, l. 20 (and p. 4, l. 31). Annual precipitation may remain similar but what about the seasonal distribution? How will temperature change affect snowfall and snowmelt, surely a very important feature of climate change in such latitudes, the water cycle and weathering?"

5    *Answer: In this part of Sweden, there is already very little snow. Otherwise is would probably be an important feature. The seasonal distribution of precipitation does change in this scenario, with less precipitation in summer in the second half of the century. The last decades the yearly precipitation increases somewhat too, about 8%. We corrected the text in the abstract with regards to seasonal distribution of precipitation and did the same in the chapter about scenarios (2.4) and in the discussion (P1L21-23, P3L5-8, P8L20-23.*

10    **Comment:** "p. 3, l. 19 (and p. 8, l.11). Why/how are base cations, Al3+ and organic acids inhibitors of weathering? Do you mean that if the concentrations of weathering products in the soil solution increase, the weathering reaction slows down as equilibrium concentrations are reached? But then aren't the weathering products being continually take away through up take, leaching or adsorption by the soil allowing weathering to proceed?"

*Answer: Yes, ions in the soil solution slow the weathering down, they don´t stop it altogether, since the weathering products*
15    *are leached, adsorbed or taken up by trees. Nonetheless, in acidified conditions the concentration of Al in the soil increases substantially compared to non-acidified conditions, and slows weathering of silicates down. Also, in the C-horizon, where water flows are slower and uptake to trees are lower, concentrations of weathering products increase and should slow weathering down to a minimum – that is probably why the C-horizon consist of less weathered parent material even though there are so much weatherable material. We corrected the text; the inhibitors base cations and aluminium are products of*
20    *the weathering (P3L19).*

**Comment:** "p. 4, l.16. There is little description of the soil type at the two sites. From the horizon abbreviations listed in Table 2, it would appear the soil are not Podzols?"

*Answer: The two soils are assessed in the SWETHRO database as transition types (developing towards podzols). O is the thin uppermost layer consisting of mostly organic material, A is greyish, but not deemed a true E zone, AB is a light brown,*
25    *B is a reddish brown and C is the parent material, which is till. We added information about the soil type in chapter 2.3 (P4L17-18).*

**Comment:** "p. 5, l.16. Furthermore, thickness of the mineral soil horizons at Hissmossa is 55 cm (and not 50 cm; Table 2) and how/why is the organic layer included where surely it is a question of organic matter decomposition rather than mineral weathering? It is not stated how stoniness was derived and if the hydraulic parameter values (Table 3) have been corrected
30    for stone content. Given the concluded importance of soil moisture, the fixed value used in PROFILE and same value for all layers (0.2) seems rather crude. The field capacity values in Table 3 also seem somewhat low – is this because of stone

content correction? How have any time related changes in organic layer thickness (and therefore soil moisture content) been taken into account?"

*Answer: The modelled soil horizon at Hissmossa is 55 cm+organic layer, but the root zone is still 50 cm+organic layer. We included the C-horizon in the modelling, even though it lies below the root zone, but we did not include it in calculations*

5 *representing the weathering in the root zone. We clarified this in the text to figure 1. The organic layers do contain some mineral soil and the modelled weathering in those layers is from the minerals, not from the organic matter. They are included in the modelling since they are important for the modelled ecosystem and thus for the soil water chemistry for the rest of the layers – for example much of the nutrient uptake takes place here, especially of nitrogen. Stoniness is estimated at the soil sampling. We added this in chapter 2.3 (P4L22-26). PROFILE, being a steady state model, can only handle fixed (in*

10 *time) values. The same soil moisture value for all horizons may be crude, but without measurements over long time periods in all soil layers, there has not been much of an alternative –and in Västra Torup's case it is in accordance with the modelling (see new figure 2). The field capacity values are calculated according to Balland (2008). Dynamic changes in layer thicknesses are not modelled by any of the models.*

**Comment:** "P. 4, l. 31: by "rainfall" you mean annual precipitation?"

15 *Answer: Yes, thanks. I have changed the text (P5L5).*

**Comment:** "p. 5, l. 6-11. The description of the scenarios is unclear, at least to me. For example, does the base scenario mean there are two thinnings and a clear cut every 70 years (and starting from the year of planting – Table 1), deposition loads constant from "todays" (which year?) levels into the future plus climate change temperature (but no change in precipitation)? What is the whole tree harvesting treatment: stems + branches or stems + branches + stumps? Is it carried out

20 every 70 years during the 1900-2100 period? It would be useful to number or letter the scenarios and refer to them in the text and, tables and figures."

*Answer: We have given the scenarios three letter abbreviations and use those throughout the text and figures, as suggested. We have clarified the scenario descriptions, both the description of the base scenario (BSC) and how the others differ from the BSC scenario (P4L31-P5L20). Yes, the BSC has forestry with a clear cut approximately every 70 years (Västra Torup:*

25 *1940, 2010, 2080. Hissmossa: 1972, 2040, 2100) and thinnings at about 25 and 45 years of plant age (the thinnings are more "on time" in the future scenario than in actual history of the sites). Deposition of SOx peaks in 1970 and decrease sharply afterwards and is in the future scenarios kept constant at 13% of the peak deposition after 2020. Deposition of N peaks in 1985 and decrease more slowly. It is kept constant at 50% of the peak deposition after 2020. Climate: Both temperature and precipitation changes in the BSC scenario, (as in the SRESA2 scenario) – the temperature increase almost*

30 *exponentially from 1900-1910 to 2090-2100 with on average about 5.9°C in winter and 3.7° in summer. Yearly precipitation changes less and without trend, but summer precipitation is lower in 2050-2100 than before and winter and spring precipitation is higher. Whole tree harvest is, in the WTH scenario of this study, stems and 60% of branches, treetops and*

*needles, but stumps are not removed. Whole tree harvest in the WTH scenario is carried out in thinning and clear cutting from the clear cutting in 2010 (Västra Torup) or 2040 (Hissmossa) and forward.*

**Comment:** "Which scenario is used for Figure 1?"

*Answer: Base scenario (BSC). We gave the scenarios abbreviations and clarified in the running text and in the figure texts*
5 *which scenario was used.*

**Comment:** "Doesn't the 70 year rotation period cover a different set of years between the two sites (20112080 vs. 2041-2100), when the climate change has changed the climate. Doesn't this explain the differences in weathering between the two sites rather than differences in soil texture (p. 5, l. 21), which anyway would also affect the soil hydraulic properties besides surface area?"

10 *Answer: Yes, the rotation periods are different for the two sites. As can be seen in figure 3 (in the previous version of the manuscript figure number 2, since we added a new figure number 2 showing soil moisture), this does not explain the difference in weathering rates between the sites, as the increase in weathering because of climate change in 20 to 30 years is a lot smaller than the difference in weathering rates between the sites. Also, if the increase in temperature in the time between the clear cuts of the two sites would have been the reason for the difference in weathering between the sites, then*
15 *Hissmossa would have had the higher weathering rates and not the lower. Also, the difference between the sites exist also in the scenarios with no climate change (NCC and BGR). See also P6L2-5. According to our calculations of soil hydraulic properties following Balland (2008), field capacity is slightly lower in Hissmossa than in Västra Torup, which means that soil moisture also is slightly lower, which should affect the weathering rates as calculated by ForSAFE, but not those calculated by PROFILE, since we used the same soil moisture value for both sites in PROFILE. And the differences in*
20 *weathering rates between the sites are as large in the PROFILE calculations as in the ForSAFE ones (see figure 1).*

**Comment:** "Why is only Mg weathering presented in Figure 2 and in Figure 4 to represent silicate mineral weathering? Wouldn't the sum of base cations be a more appropriate measure of overall silicate weathering?"

*Answer: We want to show the effect of the acidification on silicate mineral weathering rather than the total size of it, in figure 5 (previously figure 4). One of the base cations, Ca, comes in about equal amounts from weathering of apatite and of*
25 *silicates. The weathering of Ca is also relatively large. Thus, to examine the dynamics of release of base cations from only silicate weathering in our models, we chose to show one of the base cations that is only from silicate weathering, not apatite, Mg. We clarify this somewhat in the figure caption for figure 5 (previously figure 4). Figure 3 (previously figure 2) could as well show the sum of base cations, since the focus of the figure isn't difference between silicates and apatite, so we changed it to sum of base cations instead.*

30 **Comment:** "As weathering largely takes place by acid (proton) attack, why is silicate weathering decreased by acidified conditions (p. 6, l. 17) and why would apatite weathering be increased?"

*Answer: Apatite weathering is increased by more acid conditions because there are more protons in solution, and not at the same time high concentrations of something that inhibits weathering. Silicate weathering was not increased by the more acid conditions on those two sites, according to ForSAFE. Those two sites had high concentration of Al in soil solution during the acidified conditions, as Al is more soluble in acid conditions, and Al inhibits weathering of silicates in ForSAFE, but not of*

5   *apatite since there is no Al in apatite. We added a little text in the discussion (P9L11-12) to clarify this.*

**Comment:** "Doesn't the base scenario include harvesting effects besides climate change effects (p. 6, l. 29)? See also p. 8, l. 21-24. "

*Answer: Yes, but so does the NCC scenario. Forestry and acidification are equal in the NCC and the BSC scenario. Only climate differs. We clarified the scenario descriptions a bit (chapter 2.4).*

10   **Comment:** "p. 7. l. 19: by "more detailed forestry plans" do you mean timing of thinning and timing and intensity of harvesting?"

*Answer: Yes (P8L13-14). And with additional soil data, at least on soil texture, differences within a stand or a couple of nearby stands could also be modelled, so that more sensitive areas within the stand could get a less intensive forestry.*

**Comment:** "p. 8, l. 13- Isn't a matter of litter decomposition and not weathering? And I think you need to give a reference

15   that supports the statement about harvesting intensity effects of soil solution base cation concentrations. Concentrations and leaching loads may increase with whole-tree harvesting as a result of increased drainage (percolation) and disturbance of the site."

*Answer: We didn't explain this clearly enough, thank you for the question. With standard clear cutting where branches and treetops and litter are left on the site, litter decomposing after harvest lead to increases in base cation concentration,*

20   *especially of K. This can be seen in data from SWETHRO sites (krondroppsnatet.ivl.se) and in Piirainen et al. (2004). After whole tree harvesting, where branches and treetops, with most of the needles, are removed, there will be less decomposable litter on the site, which should mean that concentrations in soil water do not increase as much after WTH as after stem only harvest (as discussed in Ågren et al., 2010). The concentrations in the soil water in its turn affect the weathering rates. See P8L31-P9L2.*

25   **Comment:** "p. 8. L. 25-. The contribution of your C-horizon is small (Table 2). And if the material is less weathered, then there would be more weatherable minerals and therefore potential or weathering? Is the explanation for taking silica concentrations into account the same as mentioned above, i.e. equilibrium concentrations reached? "

*Answer: The contribution of the C-horizon is small because we only model a small portion of it. But the rates of weathering per volume in the C-horizon are equivalent to the rates in the horizons above, in the modelling. The material in the C-*

30   *horizon is less weathered since the weathering in the C-horizon is smaller than in the B-horizon, even though there is more weatherable material. The environment in the C-horizon inhibits weathering (or the C horizon would already be weathered*

*and not be a C horizon), probably because of slow water movements, so that some reactants are depleted and some products of the weathering build up and inhibit further weathering. And Si is one of the products that are building up, one that we do not model with ForSAFE and PROFILE yet. We added a little text in the discussion to clarify this, P9L33-P10L4, P10L16-21.*

5   **Comment:** "p. 9. l. 6. A paper by Starr & Lindroos (Geoderma (2006) 133: 269–280) shows this. "
**Answer:** *Thank you! We included it in the text, P10L17.*

**Comment:** "Conclusions: I appreciate the recognition of the importance of soil moisture to weathering (and decomposition) and for the reasons stated (time step). However, wouldn't a model with a daily time step be more suited to forest stand nutrient/biogeochemical cycling studies rather than modelling long-term climate change effects. "

10  **Answer:** *A daily time step is more suited for modelling of hydrology, which in its turn affects all other processes both on short term and possibly also their long term averages. The effects on long term averages might be large, if average soil moisture is affected significantly or if the dynamic shows that the timing of different processes produce for example lack of soil moisture at high temperatures or lack of nutrients when vegetation needs them. In any case, studying if the daily time step that has been implemented in another version of ForSAFE has an impact seems like a reasonable next step. We clarified*

15  *the text on model development (4.3) somewhat (P9L24).*

**Comment:** "Units: Wouldn't it be more correct to present weathering in units of moles charge rather than equivalents? And why sometimes meq m-2 yr-1 and sometimes meq m-3 y-1 (Fig. 2, 3, 4, 5). Is it somehow because the latter refer to a specific layer (of differing thicknesses) rather than to the fixed organic layer + 50 cm layer or simply a typo? "
**Answer:** *To our understanding, moles of charge and equivalents are equivalent in this case. We used weathering per area in*

20  *figure 1 to make it easy to sum up the weathering from the different layers, even though they have different thicknesses, and to show how much each layer contributes to the total weathering. But since the layers have different thicknesses, the total weathering per forest floor area for a layer is not as easily comparable to other soils in other studies as weathering per soil volume, and therefore we converted the weathering rates into meq m-3 y-1 for the text and the rest of the figures. We clarified the use of different unit in figure 1 in the caption to figure 1.*

25  **Comment:** "Table 1. Coordinates are decimal degree latitude (N) and longitude (E).
**Answer:** *Yes, thank you. We added unit in the table.*

**Comment:** "Figure 1. Title: Using which scenario and for which period of time?
**Answer:** *BSC scenario and the "future forest rotation": 2011-2080 in Västra Torup and 2041-2100 in Hissmossa. We added this in the figure caption.*

**Comment:** "Figures 2 & 6. Is it necessary to include the ForSAFE monthly values? See Fig.3.

*Answer: Since one of the objectives of the paper is to describe the dynamics of the weathering in ForSAFE it seems necessary to show the dynamics of the weathering and not only averages over time periods of different length at least in figure 2 (now figure 3) where ForSAFE weathering (=the monthly weathering values) is compared to PROFILE weathering.*

5    *We removed the monthly values from figure 6 (now figure 7).*

**Comment:** "Fig. 3. Use a circle around the years of clear cutting on the whole tree harvesting scenario and base scenario lines rather than the vertical line that intersects all scenarios. Are the years 1941, 2010 and 2080 for Västra Torup and 1973 and 2043 for Hissmossa?

*Answer: A circle would still seem to indicate all scenarios, since the average weathering is the same before the clear cut for*

10    *all three scenarios. We think that the vertical lines are more clearly visible and divides the time into the different rotation periods, but we have clarified the figure caption somewhat. The years of clear cut are 1940, 2010 and 2080 in Västra Torup and 1972, 2040 and 2101 in Hissmossa.*

**Comment:** "Fig. 4. Is each dot is a year of the 70 year rotation period? Why only layer 4 (B-horizon)?

*Answer: Each dot represents a month. L4 is the thickest layer of the root zone in Hissmossa and one of the two thickest*

15    *layers of the root zone in Västra Torup and it lies at the depth of the soil water chemistry measurements. Therefore it is the best layer for comparing modelled and measured soil water chemistry and thus the layer we look at most. Figure 4 (now figure 5) is intended to show the modelled different reaction to acidification between silicate weathering and apatite weathering and for this any layer could be used, but we chose to use L4.*

**Comment:** "Fig. 5. The base scenario includes the climate change temperature scenario and does the "constant climate

20    change scenario" refer to the last one in the list on p. 5? "

*Answer: We have now given the scenarios names and refer to them in the text and in the figures with these. The base scenario (BSC) include climate change, yes, and the other scenario used in figure 5 (now figure 6), the NCC, is the same as BSC with regards to forestry and acidifying deposition – but it has no climate change. The last scenario of the list (the BGR) has no forestry, no acidification and no climate change and is used in figure 6 (now figure 7).*

**Comments from reviewer 2**

GENERAL COMMENTS

**Comment:** "The study presents soil weathering rates for two sites in Sweden. It is concise and well written, and the subject is within the scope of SOIL."

5    *Answer: Thank you!*

**Comment:** "However, I would urge the authors to rethink the focus of the manuscript; the comparison of output from a weathering sub-model within a steady-state model to output from the same weathering sub-model in a dynamic model is somewhat trivial, and the results are unsurprising."

*Answer: ForSAFE consists of many more parts than just the weathering part and they interact with one another. Soil*
10    *moisture, for example, are not modelled at all in PROFILE and is a very important parameter for the weathering. The weathering sub model is also not exactly the same, since it is dynamic in ForSAFE. For us it was not obvious that the results would be so similar to each other. The models are different enough that one cannot just be switched to the other without studying if the newer one gives reasonable results, which is what we have done by comparing with PROFILE weathering. See also our answer to the first reviewer's general questions. We included text in the discussion clarifying how*
15    *much the models differ (P7L27-P8L7). We also expanded the second objective to refocus the paper somewhat (P2L32-33).*

**Comment:** "Similarly, it is also unsurprising that a dynamic model provides more temporal data compared to a steady-state model. The current aims of the manuscript are trivial. There is merit in presenting the dynamics of weathering; the Title and presentation could be refocused to 'Dynamic modelling of weathering rates – the benefit over steady-state modelling'."

20    *Answer: We agree with the change of title. We think there is merit also in comparing the results of a new (with regards to weathering calculations) model with a previously used model, to examine if the new model would make very different critical load or acidification sensitivity assessment than the old assessment – and if the new assessment would have been very different – how this can be explained and if it is to be believed.*

**Comment:** "Secondly, the discussion somewhat repeats the results, there are few references to literature, and overall it
25    feels more like a report."

*Answer: We have restructured the discussion somewhat, expanded it and put in more references (P7L25-P10L21).*

SPECIFIC COMMENTS

**Comment:** "P1L10. Do not mention SWETHRO in the abstract."
*Answer: We have removed it. See P1L10.*

**Comment:** "P1L20. The results / discussion of scenario's does not fit with the objectives or title of the manuscript."

*Answer: We changed the title and expanded the description of objective number two somewhat (P2L31-32).*

**Comment:** "P1L23. This is not a result of this study."

*Answer: What is not a result of the study? ForSAFE has not been used for weathering calculations before this study, but we*

5     *show that weathering rates calculated by ForSAFE are similar to weathering rates from the previously used model PROFILE and thus that ForSAFE can be used for weathering calculations too.*

**Comment:** "P2L23. Given the importance of weathering (Title / objectives), it is surprising that weathering is given limited attention in the introduction."

*Answer: We have focused on why knowing the rate of weathering is important, rather than the weathering chemistry. The*

10     *process understanding of chemical weathering, which is used in the models, has been described thoroughly in previous papers, some of which we refer to.*

**Comment:** "P2L27. I recommend that you remove the first objective and expand (refocus) the second."

*Answer: We don´t agree on removing the first objective. Weathering from the ForSAFE model has never before been investigated and compared to other, well used, ways of calculating weathering. The fact that the process is in the model*

15     *and the model gives reasonable other output does not necessarily mean that the weathering estimates are robust and useful. This study shows that the weathering estimates from ForSAFE are of the same size as estimates from the previously often used model PROFILE, and this we did not know before.*
*We expanded the description of second objective to explain the scenarios: "…scenarios, representing important ecological issues: acidification, climate change and nutrient removal through land use." (P2L31-32) The scenarios are included to*

20     *illustrate what kinds of questions dynamic modelling can help answer, using scenarios with relevant questions such as how weathering responds to climate change, changes in forestry practices and changes in acidifying deposition.*

**Comment:** "P2L30. What is the objective of the scenarios?"

*Answer: See answer to the comment above.*

**Comment:** "P3L2. What were the models applied to two forests?"

25     *Answer: The steady state model PROFILE and the dynamic model ForSAFE, described in the text in chapter 2.*

**Comment:** "P3L3. Please provide more background. For the external reader SWETHRO has no meaning or context."

*Answer: We provided some more background in chapter 2.3 (P4L3-4). There is also a reference to a paper describing SWETHRO for the reader who wishes to know details about the monitoring (on P3L2).*

**Comment:** "P3L17. Does 'factors affecting' mean sensitive parameters? Can you site previous sensitivity studies?"

*Answer: We meant parameters used to model the weathering, not only the sensitive parameters. We changed the word factors to parameters, to clarify. We cited two previous sensitivity studies in chapter 4.3. (P3L17, P9L16-17)*

**Comment:** "P4L14. There is more sand at Hissmossa (Table 2) but more less Quartz (Table 3). How is this?"

5    *Answer: Texture and total chemistry have been analysed separately and show that there are more material in the sand fraction in Hissmossa than in Västra Torup, and more Si in Västra Torup than in Hissmossa. Quartz is not only found in sand and sand does not only consist of quarts, especially not in relatively young till soils, which can explain the observed pattern. We added in the description of the sites that texture has been measured (P4L7).*

**Comment:** "P4L20. It is surprising that a fixed value is used for soil moisture given (a) that soil moisture is an 'important'

10    parameter (as noted by the authors), and that (b) the determinants of soil moisture (texture, bulk density and organic matter) are very different between both sites (as noted by the authors)."

*Answer: A steady state model cannot use a value that varies with time, so in that sense it has to be a fixed value in PROFILE. It does not have to be the same value for all sites though, but there are no measurements of soil moisture for the sites, nor are there usually any for sites that are used for PROFILE modelling. The values used are based on observation of*

15    *the vegetation on the sites, which gives a "soil moisture class", which is translated into a soil moisture value that is supposed to represent average soil moisture over a whole forest rotation, for all layers. In this study it happens to be the same for both sites. Yes, it is very crude. Modelling it, as in ForSAFE, should be a lot better. We clarified in the text in chapter 2.3 that it is a site specific value (P4L24).*

**Comment:** "P4L21. The authors need to provide better context (justification) for the scenarios."

20    *Answer: We rewrote the text explaining the scenarios in chapter 2.4 and expanded the second objective to include the scenarios better (P2L31-32, P4L32-P5L20).*

**Comment:** "P5L5. The list of scenarios suggests a study objective different than that presented."

*Answer: We expanded the second objective somewhat to include the scenarios better (P2L31-32).*

**Comment:** "P5L25. Given the importance of soil moisture, why are these data not shown (Figure or Table)"

25    *Answer: Good idea, data on modelled soil moisture is included as a new figure 2.*

**Comment:** "P5 Figure 1. The difference between PROFILE and ForSAFE in L4 (and L5) at Hissmossa needs more quantitative explanation / support. It might be soil moisture but this is not clearly shown."

*Answer: In Hissmossa, there is a very strong relationship between difference in soil moisture in the two models and*

*difference in weathering in the two models. This can be more easily seen now when the soil moisture is shown in a new*

*figure 2. In Västra Torup the differences between soil moisture in the two models are small, except in the organic layer, and*

*differences between weathering rates between the two models are also small, except in the organic layer. There are other*

*differences between the models that affect the difference in weathering rates too, but soil moisture has a large role. See*

5   *figure below.*

[Figure]

*Figure 1. Difference in calculated weathering between ForSAFE and PROFILE vs difference in soil moisture between ForSAFE*

*and PROFILE, in percent. The largest relative differences are in the organic layers of both sites, where weathering is very*

*low.*

10   **Comment:** "P6 Figures 2 to 6. Many of the figures forces on magnesium or calcium but the sum of base cations is the focus

of the text (primarily)."

**Answer:** *We changed figure 2 (now figure 3) to two diagrams that show the sum of base cations instead of Mg. Figure 3*

*(now figure 4) shows sum of base cations. Figure 4 (now 5) and 6 (now 7) show Mg and P, since the rates of weathering of*

*these two react in the opposite way to acidification, which was the focus of figure 4 (now 5) and the reason behind some of*

15   *the difference between the base scenario and the background scenario shown in figure 6 (now 7). Figure 5 (now 6) shows*

*Ca as an example of one of the base cations.*

**Comment:** "P7L10. The discussion has notably few citations. . . it is a discussion?"

**Answer:** *We have restructured the discussion and put in some more references. See answer to the general question*

*regarding the discussion.*

**Comment:** "P9L15–L19. These are not surprising conclusions (and more-or-less were previously known)."

*Answer: That two different models give comparable results needs to verified before using the new, previously untested model (with regards to this process), which is what we have done. For us, the results were not at all self-evident. We changed the wording of the conclusions somewhat, since yes, it is quite self-evident that the ForSAFE model provides more results, but not necessarily that they are useful (P10L26).*

**Comment:** "Table 3. It appears that surface area is estimated for Clay, Silt and Sand. How are areas for the O horizon estimated?"

*Answer: For the mineral part of the soils in the organic layers, mineralogies and texture analyses from the second layers are used, since there are no texture analyses for the organic layers and the total chemistry analysis of them include the ash of the organic part and thus aren't useful for calculating mineralogy. The surface areas of the mineral part of the organic layers are calculated from the texture in the second layers, but adjusted for how much less mineral matter there is in the organic layers (about 10% of the matter). We clarified this in the text in chapter 2.3 (P4L22-24).*

**Manuscript:**

[revised manuscript text omitted]
 (m) | Bulk density (kg m$^{-3}$) | OM (% of DW) | Stoniness (%) | Clay | Silt | Sand | pH H$_2$O | Al | H | Na | K | Mg | Ca | CEC (µeq g$^{-1}$) | BS (%) | Tot-C | Tot-N |
|---|---|---|---|---|---|---|---|---|---|---|---|---|---|---|---|---|---|---|
| | | | | | (% of mineral soil) | | | | Exchangeable ions (µeq g$^{-1}$) | | | | | | | | (g (kg DW)$^{-1}$) | |
| **Västra Torup** | | | | | | | | | | | | | | | | | | |
| O | 0.05 | 181 | 87 | 0 | | | | 4.0 | 29 | 84.5 | <0.1 | 13.0 | 27.9 | 50.1 | 205 | 43.0 | 543 | 20.9 |
| A | 0.06 | 959 | 6 | 20 | 5 | 27 | 68 | 4.1 | 31 | 16.5 | <0.1 | 1.0 | 0.7 | 0.8 | 50 | 4.9 | 34 | 2.0 |
| AB | 0.20 | 1062 | 5 | 20 | 5 | 31 | 64 | 4.6 | 27 | 6.1 | <0.1 | 0.6 | 0.4 | 1.2 | 36 | 6.4 | 25 | 1.7 |
| B | 0.20 | 1279 | 4 | 20 | 3 | 21 | 76 | 4.8 | 16 | 1.3 | <0.1 | 0.4 | 0.1 | 0.6 | 18 | 6.5 | 18 | 1.3 |
| C | 0.04 | 1446 | 2 | 20 | 0 | 17 | 83 | 4.9 | 13 | 4.8 | <0.1 | 0.4 | 0.1 | 0.5 | 19 | 5.0 | 8 | 0.6 |
| **Hissmossa** | | | | | | | | | | | | | | | | | | |
| O | 0.05 | 394 | 65 | 0 | | | | 3.5 | 45 | 63.5 | 3.7 | 8.1 | 21.4 | 26.4 | 164 | 36.8 | 391 | 19.8 |
| A | 0.13 | 909 | 8 | 10 | 0 | 5 | 91 | 3.8 | 36 | 15.3 | 0.6 | 1.4 | 3.8 | 2.9 | 60 | 17.6 | 46 | 3.0 |
| AB | 0.10 | 1075 | 8 | 10 | 1 | 8 | 89 | 4.6 | 27 | 4.6 | 0.4 | 0.8 | 2.5 | 2.3 | 37 | 17.8 | 38 | 3.0 |
| B | 0.28 | 1276 | 3 | 10 | 0 | 9 | 88 | 4.5 | 12 | 0.3 | 0.4 | 0.7 | 2.3 | 2.3 | 17 | 34.2 | 17 | 2.2 |
| C | 0.04 | 1316 | 3 | 10 | 0 | 8 | 88 | 4.7 | 11 | 0.7 | 0.4 | 0.7 | 2.4 | 2.3 | 17 | 36.1 | 14 | 2.0 |

[revised manuscript text omitted]